# Regulation of posterior body and epidermal morphogenesis in zebrafish by localized Yap1 and Wwtr1

David Kimelman[1]*, Natalie L Smith[1†], Jason Kuan Han Lai[2†], Didier YR Stainier[2]

[1]Department of Biochemistry, University of Washington, Seattle, United States; [2]Department of Developmental Genetics, Max Planck Institute for Heart and Lung Research, Bad Nauheim, Germany

**Abstract** The vertebrate embryo undergoes a series of dramatic morphological changes as the body extends to form the complete anterior-posterior axis during the somite-forming stages. The molecular mechanisms regulating these complex processes are still largely unknown. We show that the Hippo pathway transcriptional coactivators Yap1 and Wwtr1 are specifically localized to the presumptive epidermis and notochord, and play a critical and unexpected role in posterior body extension by regulating Fibronectin assembly underneath the presumptive epidermis and surrounding the notochord. We further find that Yap1 and Wwtr1, also via Fibronectin, have an essential role in the epidermal morphogenesis necessary to form the initial dorsal and ventral fins, a process previously thought to involve bending of an epithelial sheet, but which we now show involves concerted active cell movement. Our results reveal how the Hippo pathway transcriptional program, localized to two specific tissues, acts to control essential morphological events in the vertebrate embryo.

DOI: https://doi.org/10.7554/eLife.31065.001

*For correspondence:
kimelman@uw.edu

†These authors contributed equally to this work

## Introduction

A key step in the development of the vertebrate embryonic body is the change from the roughly spherical-shaped embryo at the end of gastrulation to the elongated body present at the end of the somite-forming stages when the initial anterior-posterior body plan is fully established (reviewed in *Bénazéraf and Pourquié, 2013*; *Henrique et al., 2015*; *Kimelman, 2016*; *Wilson et al., 2009*). This process involves complex morphogenetic changes as cells migrate out of a bipotential neuromesodermal progenitor population located at the most posterior end of the embryo and into the presomitic mesoderm and posterior neural tube (*Bénazéraf et al., 2010*; *Goto et al., 2017*; *Kanki and Ho, 1997*; *Lawton et al., 2013*; reviewed in *McMillen and Holley, 2015*; *Serwane et al., 2017*; *Steventon et al., 2016*).

The factors and mechanisms regulating these morphogenetic events are still largely unknown, but are an active area of research. The mesodermal transcription factors Tbx6/16 and Mesogenin play an essential role in this process as mutants in these genes fail to form the posterior body axis normally (*Chalamalasetty et al., 2014*; *Chapman and Papaioannou, 1998*; *Fior et al., 2012*; *Kimmel et al., 1989*; *Manning and Kimelman, 2015*; *Nowotschin et al., 2012*; *Yabe and Takada, 2012*; *Yoon and Wold, 2000*), although the relative importance of each factor is species specific (*Kimelman, 2016*; *Nowotschin et al., 2012*). The signaling factors Wnt and Fgf also play essential roles by controlling various aspects of cell movement at the posterior end of the embryo (*Bénazéraf et al., 2010*; *Goto et al., 2017*; *Lawton et al., 2013*; *Steventon et al., 2016*).

An additional factor that was shown to have a major role in overall early vertebrate body morphogenesis is the Hippo pathway transcription factor Yap1, a key mechanotransduction sensor whose

activity is regulated both through protein degradation and nucleocytoplasmic shuttling (*Piccolo et al., 2014*). In this study (*Porazinski et al., 2015*), a medaka *yap1* mutant (*hirame*) was shown to have a severely distorted body form that was described as being a flattened embryo, although the specific role of Yap1 in forming the posterior body during somitogenesis was not examined. Yap1 was thus proposed to regulate tissue tension throughout the medaka embryo, thereby opposing the forces of gravity. Based on additional studies using 3D spheroids of human retinal epithelial cells, Yap1 was proposed to act by regulating the transcription of one or more Rho GTPase activating protein genes (*arhgaps*), although these genes were not shown to be functional targets of Yap1 in the medaka embryo (*Porazinski et al., 2015*).

Using zebrafish embryos, we show here that simultaneous loss of Yap1 and its paralog Wwtr1 (previously called Taz, *Hilman and Gat, 2011*; *Piccolo et al., 2014*), causes a severe defect in the elongation of the embryonic body during the mid-late somitogenesis stages. Surprisingly, we find that Yap1 and Wwtr1 are not ubiquitously expressed throughout the embryo but are specifically localized to the presumptive epidermis and notochord. Analysis of transcriptional changes in $yap1^{-/-}$; $wwtr1^{-/-}$ double mutants reveals downregulation of a spectrum of genes expressed within the presumptive epidermis and notochord, but with little or no effect on the expression of *arhgaps*. Examination of the presumptive epidermis in *yap1;wwtr1* double mutants also revealed profound defects, leading to a new understanding of the basis of the initial steps in the formation of the median fin fold, the precursor to the dorsal and ventral unpaired fins. Importantly, we discovered that the *yap1*; *wwtr1* double mutants have aberrant Fibronectin (Fn) deposition, and show that interfering with Fn function results in embryos that appear similar to the *yap1;wwtr1* double mutants, with inhibition of posterior elongation and a failure of presumptive epidermal cells to move into the nascent fin fold. We propose that Yap1 and Wwtr1 regulate transcription programs specifically within presumptive epidermal and notochord cells, allowing them to adhere to the surrounding tissues and form the extracellular matrix that is necessary for both posterior body elongation and presumptive epidermal cell migration to produce the median fin fold.

## Results

### *yap1;wwtr1* double mutants exhibit severe posterior body defects

We generated new mutant alleles of *yap1* and *wwtr1* using CRISPR (*Lai et al., 2017*). $yap1^{bns19}$ contains a 41 bp deletion in exon 1, which encodes part of the TEAD binding domain, whereas $wwtr1^{bns35}$ contains a 29 bp insertion in exon 2, which encodes part of the WW domain (*Figure 1A*). Both frameshift mutations are predicted to lead to truncated proteins (p.Ile39Argfs*72 and p. Pro145Glnfs*15, respectively). *yap1;wwtr1* double homozygous mutants ($yap1^{-/-};wwtr1^{-/-}$) exhibit a severe defect in forming the posterior body, first clearly visible at the 15–16 somite stage (*Figure 1B* and *Video 1*). Although not studied, the same defect was observed previously in zebrafish double mutant embryos with different *yap1* and *wwtr1* mutations (Figure 1G in *Miesfeld et al., 2015*; Supplemental Figure 6I in *Nakajima et al., 2017*), demonstrating that the defects we saw were not due to the particular alleles we had identified. Prior to the 15–16 somite stage we did not observe any differences among the embryos during gastrulation or early somitogenesis, indicating that the role of zygotic Yap1/Wwtr1 becomes essential as the tail begins to form. *yap1;wwtr1* double mutants formed normally proportioned somites, although the somites never obtained the chevron shape seen in wild-type or sibling embryos (*Figure 1C*). The body was not flattened by gravity (*Figure 1E, F*) as described for medaka *yap1* mutants (*Porazinski et al., 2015*). We also observed that among the embryos that arose from crosses of $yap1^{+/-};wwtr1^{+/-}$ adults, a proportion showed less severe defects after the embryonic stages, including milder posterior defects (*Figure 1D*). When these embryos were genotyped (n = 15) they were all found to be $yap1^{-/-};wwtr1^{+/-}$, showing that while one copy of *yap1* alone is sufficient for posterior body formation, one copy of *wwtr1* is not sufficient, in the context of the absence of its paralogue.

### Yap1 and Wwtr1 are localized to the notochord and presumptive epidermis

During the somitogenesis stages, separate groups have reported somewhat different distributions of *yap1* mRNA in early zebrafish embryos (*He et al., 2015*; *Jiang et al., 2009*; *Loh et al., 2014*) and

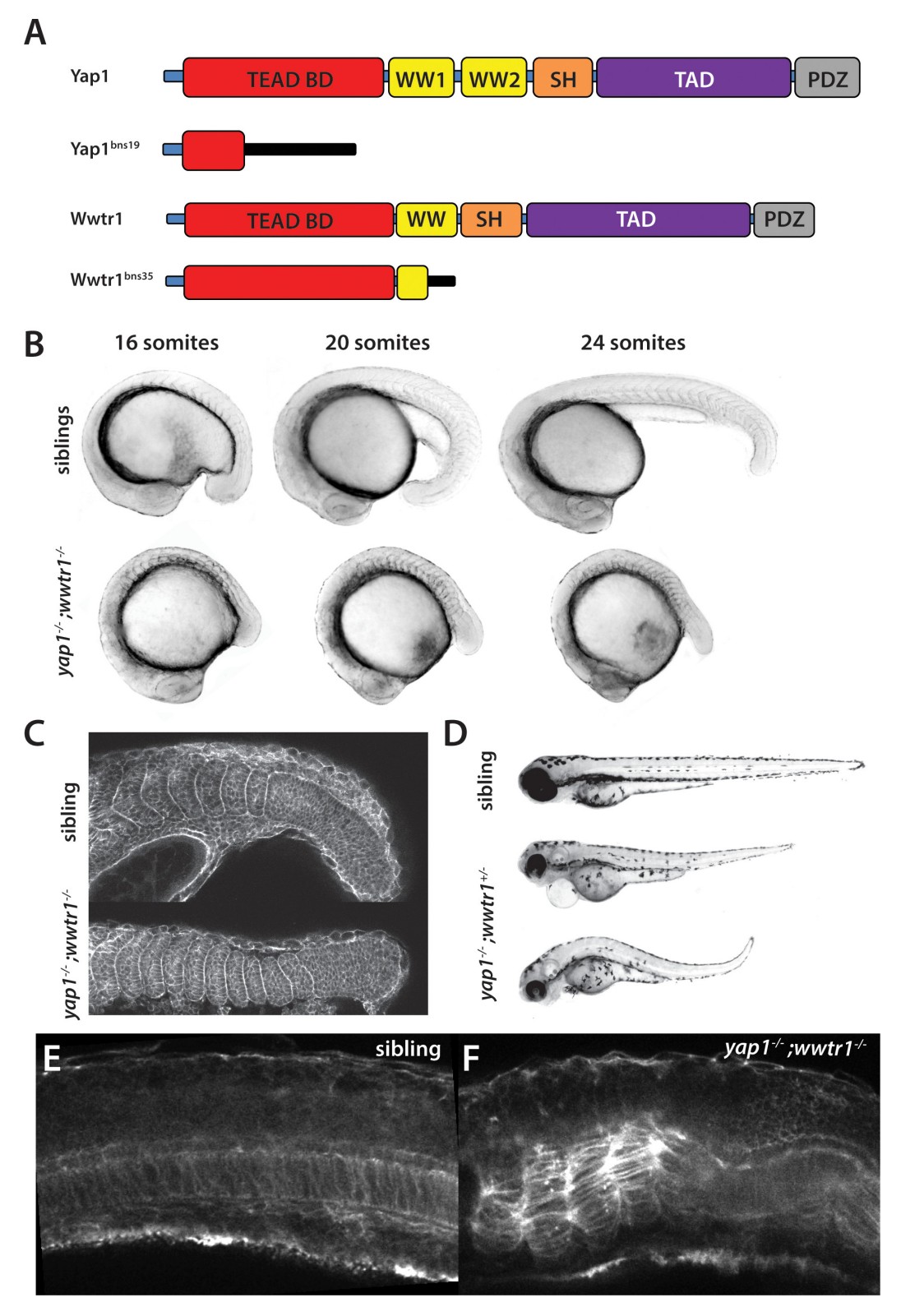

**Figure 1.** *yap1;wwtr1* double mutants exhibit severely altered posterior development. (**A**) Schematic showing the two mutants used in this work. TEAD BD is the TEAD binding domain, TAD is the Transcriptional Activation Domain, and WW, SH and PDZ are protein interaction domains. The black bar indicates new sequence following the frame shift caused by the mutation. (**B**) Time course showing the development of the posterior body defect in *yap1;wwtr1* double mutants. The same sibling and double mutant embryos are photographed at all three stages. (**C**) Phalloidin staining shows that the
*Figure 1 continued on next page*

*Figure 1 continued*

somites form in the mutants but never acquire the chevron shape found in wild-type embryos. 24-somite stage (21 hpf) embryos, anterior to the left. (D) Embryos with a *yap1^-/-;wwtr1^+/-* genotype show a milder posterior body defect at 72 hpf (n = 15). Pericardial edema is also variably present. (E,F) Midline confocal sections of the trunk with somite 1 on the left side of a sibling (E) and a *yap1;wwtr1* double mutant embryo (F) at the 24-somite stage.
DOI: https://doi.org/10.7554/eLife.31065.002

our data not shown). In medaka, only the temporal expression of *yap1* using PCR analysis was described (*Porazinski et al., 2015*). The expression of *wwtr1* mRNA in neither system has been reported during the somitogenesis stages. Since the protein distribution of these factors, whose levels and cellular localization are controlled post-translationally (*Piccolo et al., 2014*), have not been analyzed during somitogenesis in any vertebrate embryo, we examined the expression of Yap1 and Wwtr1 in zebrafish at mid-somitogenesis, when the mutant phenotype first becomes clear, in the posterior of the embryo. Surprisingly, Yap1 was strongly expressed in the presumptive epidermis at the 18-somite stage, as well as along the axis (*Figure 2A*). In both cases, the staining was nuclear, which is indicative of active Yap1 (*Piccolo et al., 2014*). The presumptive epidermal expression of Yap1 was confirmed with fluorescent in situ hybridization (FISH) using probes for the previously characterized presumptive epidermal markers *keratin4* and *keratin8* (*Figure 2—figure supplement 1*). The expression of Yap1 protein in the notochord was ascertained using FISH with a probe for the notochord marker *no tail*, also called *ta1* (*Schulte-Merker et al., 1994*, *Figure 2B*). Importantly, *yap1* mutants did not exhibit any Yap1 antibody staining, demonstrating that the antibody is specific to Yap1 (*Figure 2C*). Similarly, Wwtr1 was also localized to the presumptive epidermis and notochord (*Figure 2—figure supplement 2*), in keeping with the overlapping role of these two factors (*Zanconato et al., 2015*). Based on these findings we also examined the expression of Yap1 in medaka to determine if zebrafish and medaka were significantly different. As in zebrafish, medaka Yap1 is also predominantly expressed in the presumptive epidermis and notochord in the posterior (*Figure 2—figure supplement 2*). Altogether, these data suggest that Yap1 and Wwtr1 are not acting ubiquitously throughout the embryo but function in localized domains of the posterior body to control morphogenesis.

## Identification of target genes regulated by Yap1 and Wwtr1

In order to understand precisely how Yap1 and Wwtr1 are regulating posterior morphogenesis, the tailbud and presomitic mesoderm up to the third newest somite (S-III) were manually dissected at the 16–18 somite stage from embryos collected from a cross of *yap1^+/-;wwtr1^+/-* adults, and which had been separated by phenotype into siblings and *yap1;wwtr1* double mutant embryos, and then subjected to RNA-seq analysis (*Supplementary file 1*). As shown in *Table 1*, most of the top hits have not been previously analyzed, with only four having been localized to the presumptive epidermis or notochord during the somitogenesis stages (data from ZFIN.org). We therefore produced in situ hybridization probes to the remaining genes shown in *Table 1* and found that their expression all localized to the presumptive epidermis or notochord at the 18-somite stage (*Table 1*). For example, *ecrg4b* is expressed in the presumptive epidermis at this stage, but is not detected in *yap1;wwtr1* double mutants, confirming the RNA-seq results (*Figure 3A,B*). Similarly, *wu: fc23c09*, a zebrafish *podocan* family member, is expressed throughout the notochord in sibling embryos but is mostly missing from the posterior notochord in *yap1;wwtr1* double mutants (*Figure 3—figure supplement 1A,B*), which is within the region that was isolated for the RNA-seq analysis.

We also used Gene Ontology (GO) analysis to further analyze our RNA-seq results. In accord with the primary targets of Yap1/Wwtr1 being

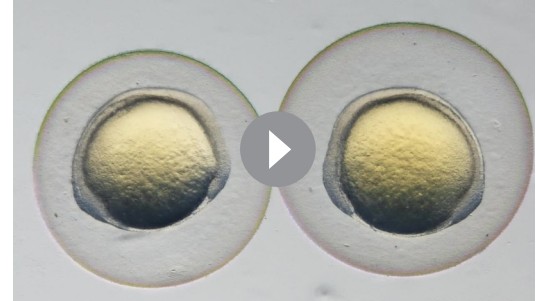

**Video 1.** Comparison of sibling (left) and *yap1;wwtr1* double mutant (right) embryos. The movie begins at the 3-somite stage.
DOI: https://doi.org/10.7554/eLife.31065.003

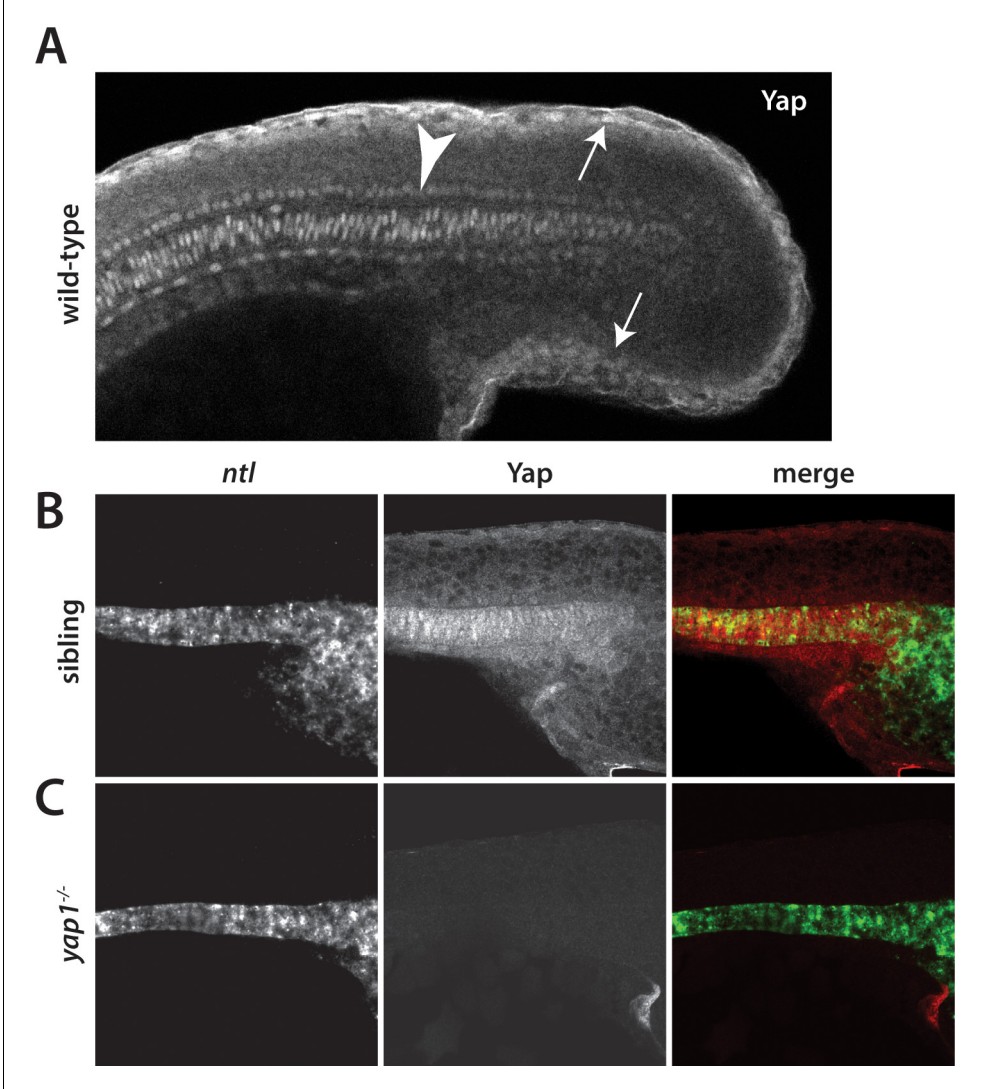

**Figure 2.** Yap1 is localized to the presumptive epidermis and axis. (**A**) Yap1 protein is expressed in the presumptive epidermis (arrows) and midline (arrowhead) of an 18-somite stage embryo. (**B**) Co-stain with the notochord marker *no tail* (*ntl*), in green, using FISH demonstrates Yap1 expression, in red, in the notochord. (**C**) No reactivity with the Yap1 antibody was observed in *yap1*⁻/⁻ mutant embryos. Note that the longer Proteinase K digestion used in the standard FISH protocol reduced the presumptive epidermal Yap1 staining. All embryos are at 18 somites with anterior to the left.

DOI: https://doi.org/10.7554/eLife.31065.004

The following figure supplements are available for figure 2:

**Figure supplement 1.** Yap1 localizes to the presumptive epidermis.
DOI: https://doi.org/10.7554/eLife.31065.005

**Figure supplement 2.** Wwtr1 and medaka Yap1 localize to the presumptive epidermis and notochord.
DOI: https://doi.org/10.7554/eLife.31065.006

expressed in the presumptive epidermis, categorization by Biological Process revealed genes expressed in either the epidermis or the fin, which is primarily an epidermal structure (*Figure 4A*). Notochord did not show up as statistically significant in this analysis both because only a small number of Yap1/Wwtr1 targets are expressed in the notochord (*Table 1*) and because several genes expressed in the notochord in zebrafish embryos are not identified as being expressed in the notochord in the GO database. Interestingly, the major targets identified using Kegg Pathway analysis comprised genes involved in the cytoskeleton, or in the interactions of cells with their environment

**Table 1.** Gene expression changes in *yap1;wwtr1* double mutants

| ENSEMBL gene ID | ENSEMBL gene name | Protein type | Fold-change (decrease) | Expression domain |
|---|---|---|---|---|
| ENSDARG00000088717 | *ecrg4b* | Augurin | 51.1 | Presumptive epidermis* |
| ENSDARG00000086539 | *CABZ01115881.1* | SH2D3A | 12.1 | Presumptive epidermis* |
| ENSDARG00000061948 | *amotl2b* | Amotl | 11.7 | Presumptive epidermis* |
| ENSDARG00000019365 | *zgc:110712* | Keratin 14/17 | 11.1 | Presumptive epidermis* |
| ENSDARG00000056627 | *cxcl14* | Chemokine (C-X-C motif) | 10.5 | Presumptive epidermis |
| ENSDARG00000088002 | *wu:fc23c09* | Podocan | 8.3 | Notochord* |
| ENSDARG00000098058 | *im:7150988* | GAPR-1 | 7.9 | Presumptive epidermis |
| ENSDARG00000094752 | *rpe65b* | Carotenoid oxygenase | 7.8 | Presumptive epidermis* |
| ENSDARG00000025254 | *s100a10b* | Calcium binding | 7.3 | Presumptive epidermis* |
| ENSDARG00000074002 | *slc6a11a* | Solute carrier | 6.8 | Presumptive epidermis* |
| ENSDARG00000101423 | *cyp2y3* | Cytochrome P450 | 6.3 | Presumptive epidermis |
| ENSDARG00000023062 | *cyr61* | CCN family ECM Protein | 6.3 | Notochord |

*Our data; others are from ZFIN.org

DOI: https://doi.org/10.7554/eLife.31065.007

(*Figure 4B*). Finally, we examined genes previously identified as Hippo pathway transcriptional targets in large-scale screens in other systems (*Zhang et al., 2009*; *Zhao et al., 2008*). The top 10 targets are shown in *Table 2*, and all except one of these are expressed in the presumptive epidermis or notochord. The one exception is *nuak1a*, which is weakly expressed in the ventral mesoderm and thus appears to be another tissue regulated by Yap1/Wwtr1, albeit for a very limited set of target genes. Because the genes expressed in *Table 2* have more modest fold-changes, we also validated that their expression decreased using quantitative PCR (*Figure 4—figure supplement 1*).

To further understand the regulation of these genes by Yap1 and Wwtr1, we made use of two transgenic lines that allow temporal regulation of Yap1/Wwtr1, *Tg(hsp70:RFP-DNyap)*, which expresses a dominant-negative form of Yap1 under heat shock control, and *Tg(hsp70:RFP-CAyap)*, which expresses a constitutively active Yap1 under heat shock control (*Mateus et al., 2015*). These lines are referred to here as *HS:DN-yap* and *HS:CA-yap*, respectively. Embryos were heat shocked at the 4-somite stage and then fixed at 18 somites. Inhibition of Yap1/Wwtr1 function with *HS:DN-yap* caused a downregulation of *ecrg4b* expression, although not as strongly as observed in the *yap1; wwtr1* double mutants (*Figure 3C,D*). Intriguingly, enhancement of Yap1/Wwtr1 levels with *HS:CA-yap* caused an upregulation of *ecrg4b*, indicating that Yap1/Wwtr1 are not working to activate transcription to maximum levels (*Figure 3E,F*). In contrast, while *HS:DN-yap* caused an inhibition of *wu: fc23c09* expression in the posterior notochord similar to that observed in the mutants, *HS:CA-yap* did not upregulate *wu:fc23c09*, indicating that Yap1/Wwtr1 already functions at maximal levels in this tissue, or at least for this target (*Figure 3—figure supplement 1C,E*). Interestingly, the increased expression of *ecrg4b* caused by CA-yap was restricted to the presumptive epidermis (*Figure 3G,H*), and the absence of *wu:fc23c09* expression caused by DN-yap was restricted to the posterior notochord (*Figure 3—figure supplement 1D*). These observations indicate that Yap1/Wwtr1 must work with tissue-specific factors to control gene expression.

Since the analysis of a *yap1* mutant in medaka proposed that *arhgaps* were key targets of Yap1 (*Porazinski et al., 2015*), we examined *arhgaps* expression in our RNA-seq data. Of the many *arhgaps* found in zebrafish, only two weakly expressed *arhgaps* (based on the RNA-seq data), *arhgap12a* and *arhgap27*, were observed to change between siblings and mutants, with *arhgap12a* down 1.8-fold in the mutants and *arhgap27* down 1.9-fold (*Supplementary file 1*). However, we also found that many presumptive epidermal genes were decreased approximately 2-fold in the mutants, including all of the abundant *keratins*. For example, *krt4* was down 2.4-fold, *krt8* was down 2.7-fold, and *krt18* was down 1.8-fold. We also examined the expression of these two *arhgap* genes by in situ hybridization to determine where they were expressed and to look for possible changes in expression in the mutants. We found that both *arhgaps* are expressed in the presumptive epidermis, and

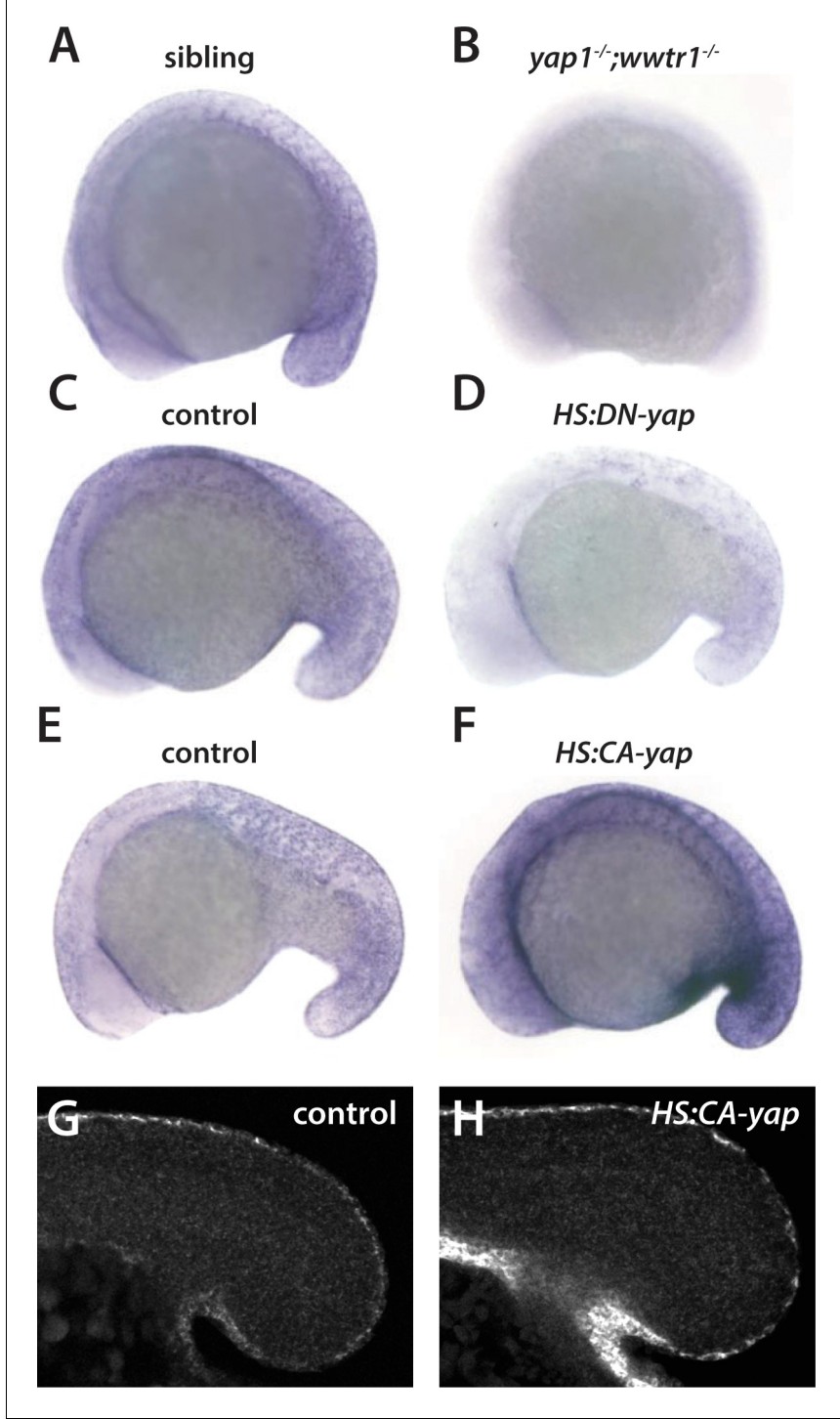

**Figure 3.** Regulation of *ecrg4b* expression by Yap1 and Wwtr1. (**A,B**) *ecrg4b* is expressed in the presumptive epidermis of sibling embryos (**A**) but is not expressed in *yap1;wwtr1* double mutant embryos (**B**). (**C–F**) Transgenic embryos obtained from an outcross of hemizygous *HS:DN-yap* or *HS:CA-yap* adults were heat shocked at the 4-somite stage and then raised to 18 somites. The genotype of the embryos was determined by PCR after the in situ hybridization to be either transgenic or non-transgenic control. Expression of DN-yap decreased *ecrg4b* expression relative to non-transgenic controls (C,D, n = 6 for each) whereas CA-yap enhanced *ecrg4b* expression (E,F, n = 4 for each). (**G,H**) Embryos analyzed using FISH probe to *ecrg4b* demonstrates that CA-yap increases expression of *ecrg4b* only in the epidermis (n = 4 for each). All embryos are at the 18-somite stage.
DOI: https://doi.org/10.7554/eLife.31065.008

*Figure 3 continued on next page*

*Figure 3 continued*

The following figure supplements are available for figure 3:

**Figure supplement 1.** Regulation of *wu:fc23c0* by Yap1 and Wwtr1.
DOI: https://doi.org/10.7554/eLife.31065.009

**Figure supplement 2.** Yap1 and Wwtr1 do not regulate *arhgap12* and *arhgap27* expression.
DOI: https://doi.org/10.7554/eLife.31065.010

found no obvious spatial difference in their expression between siblings and mutants (*Figure 3—figure supplement 2*). One *arhgap* gene is expressed in the posterior notochord, *arhgap17a* (data from ZFIN.org), but this gene showed no change between siblings and mutants in our RNA-seq data. Our data suggest that *arhgaps* are not the key targets of Yap1/Wwtr1 in regulating posterior body morphogenesis in zebrafish (see Discussion), and instead identify a number of genes whose expression is highly regulated by these transcription factors in either the presumptive epidermis or notochord, which are the same tissues in which we found Yap1 and Wwtr1 expression.

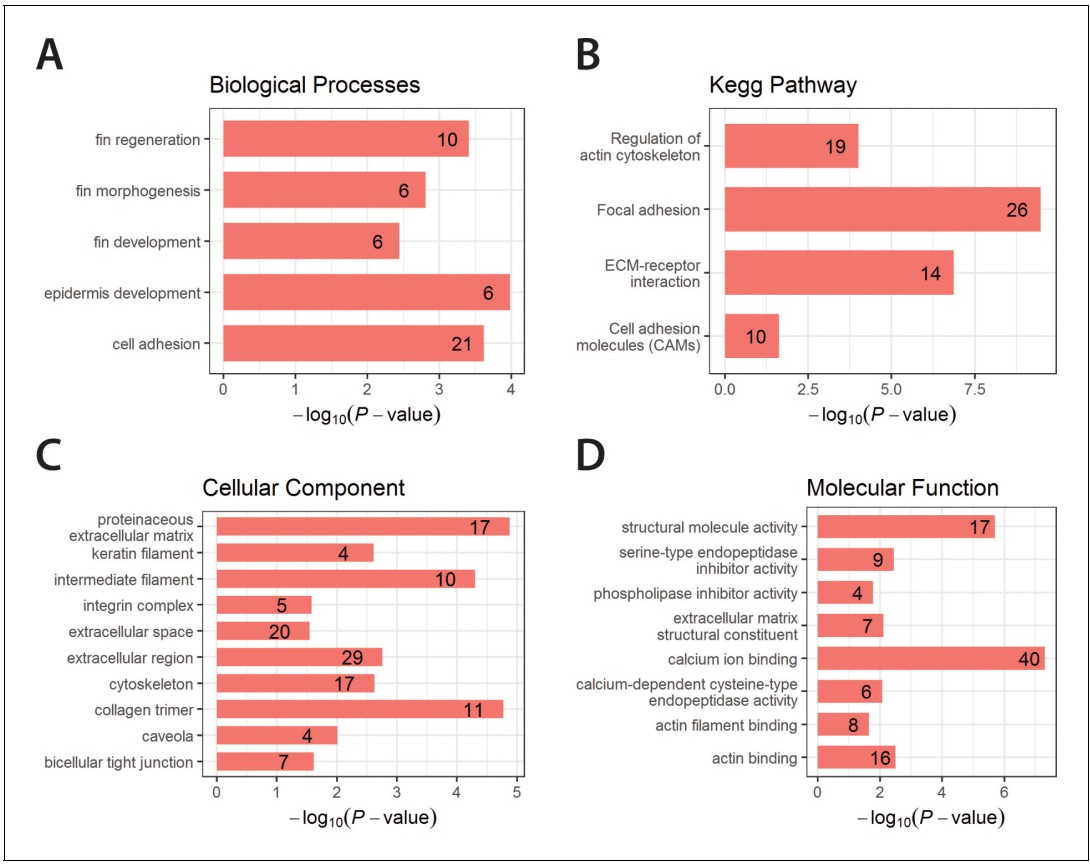

**Figure 4.** Enriched GO terms and Kegg Pathway for downregulated genes in *yap1;wwtr1* double mutants. Bar graphs showing the log-transformed adjusted *P*-values of enriched GO terms categorized under Biological Processes (**A**), Cellular Component (**C**) and Molecular Function (**D**), and for the Kegg Pathway (**B**). Number of genes in the enriched terms is shown in respective bar graphs.
DOI: https://doi.org/10.7554/eLife.31065.011

The following figure supplements are available for figure 4:

**Figure supplement 1.** Real-time PCR (qPCR) assays validate downregulation of known Yap1/Wwtr1 target genes.
DOI: https://doi.org/10.7554/eLife.31065.012

**Figure supplement 2.** Morphogenetic defects in *noto* mutants.
DOI: https://doi.org/10.7554/eLife.31065.013

**Table 2.** Regulation of previously identified Yap1/Wwtr1 target genes[†]

| ENSEMBL gene ID | ENSEMBL gene name | Protein type | Fold-change (decrease) | Expression domain |
|---|---|---|---|---|
| ENSDARG00000061948 | amotl2b | Amotl | 11.7 | Presumptive epidermis* |
| ENSDARG00000023062 | cyr61 | CCN family ECM Protein | 6.2 | Notochord |
| ENSDARG00000012066 | dcn | Decorin | 4.4 | Presumptive epidermis |
| ENSDARG00000052783 | cdc42ep3 | Cdc42 Effector Protein | 4.1 | Presumptive epidermis |
| ENSDARG00000006603 | csrp1a | Cys and Gly Rich Protein | 4.0 | Notochord |
| ENSDARG00000037476 | sorbs3 | Scaffold protein | 3.6 | Presumptive epidermis |
| ENSDARG00000042934 | ctgfa | CCN family ECM Protein | 3.5 | Notochord |
| ENSDARG00000035809 | col1a1b | Collagen | 3.5 | Presumptive epidermis |
| ENSDARG00000020086 | nuak1a | Amp-Activated Protein Kinase | 3.4 | Ventral mesoderm* |
| ENSDARG00000060610 | pcdh7b | Protocadherin | 3.4 | Presumptive epidermis |

*Our data; others are from ZFIN.org
[†]From *Zhao et al. (2008)* and *Zhang et al. (2009)*
DOI: https://doi.org/10.7554/eLife.31065.014

## Notochord-less embryos show a weak posterior body defect

The localization of Yap1 and Wwtr1 to the presumptive epidermis and notochord, together with the changes in gene expression in these tissues, indicated that defects in one or both tissues were the source of the posterior morphogenetic defects. Since the notochord cells undergo active morphogenetic movements during somitogenesis in many species (*Jiang and Smith, 2007*; *Keller et al., 2000*; *Lee and Anderson, 2008*; *McMillen and Holley, 2015*), it seemed possibly to be the key target tissue for Yap1/Wwtr1 regulated posterior morphogenesis. A zebrafish mutant that lacks all notochord formation, *noto1/floating head*, was previously identified in large-scale genetic screens, and shown to have a shortened body axis at 24 hpf (30 somite stage, *Odenthal et al., 1996*; *Stemple et al., 1996*; *Talbot et al., 1995*). Since posterior morphogenesis at earlier stages had not been characterized, we obtained the original *noto1[n1]* mutant line (*Talbot et al., 1995*) and examined the formation of the posterior body. At the 16-somite stage there was a minor but recognizable difference between *noto* mutants and their siblings, and this difference became more pronounced as embryos developed to the 24-somite stage, paralleling the time course of alterations in the *yap1;wwtr1* double mutants (*Figure 4—figure supplement 2*). However, since the defects in *noto* mutants were much less severe than those in *yap1;wwtr1* double mutants, our results suggested that alterations in the notochord are only a small part of the overall *yap1;wwtr1* double mutant phenotype.

## Alterations in the presumptive epidermis in *yap1;wwtr1* double mutants

To determine whether the presumptive epidermis was altered in *yap1;wwtr1* double mutants we first examined the expression of cytokeratins using two antibodies, panKr1-8, which recognizes a broad spectrum of cytokeratins, and 79.14, which recognizes a more limited subset (*Conrad et al., 1998*). We found that PanKr1-8 labeled the entire presumptive epidermis, whereas 79.14 only labeled the outermost cells (*Figure 5A,C*). With both antibodies we observed that both the dorsal and ventral midlines at the 24-somite stage were 1–2 cell layers thick in mutants, whereas in siblings they were several layers thick (*Figure 5A–D*).

Because the cytokeratin antibodies do not work well at earlier stages, we turned to an antibody to the transcription factor Tp63, which marks the formation of the basal layer of the nascent presumptive epidermis from mid-gastrula stages onward (*Bakkers et al., 2002*; *Lee and Kimelman, 2002*) in order to examine the presumptive epidermis during mid-somitogenesis stages. From the 18-somite stage to the 24-somite stage (a period of 3 hr), the embryo undergoes a rapid change in the thickness of the presumptive epidermis along the dorsal and ventral midlines to form the nascent median fin fold (MFF) that ultimately gives rise to the dorsal and ventral unpaired fins (*Figure 6A*), as was previously observed in histological sections (*Abe et al., 2007*; *Dane and Tucker, 1985*).

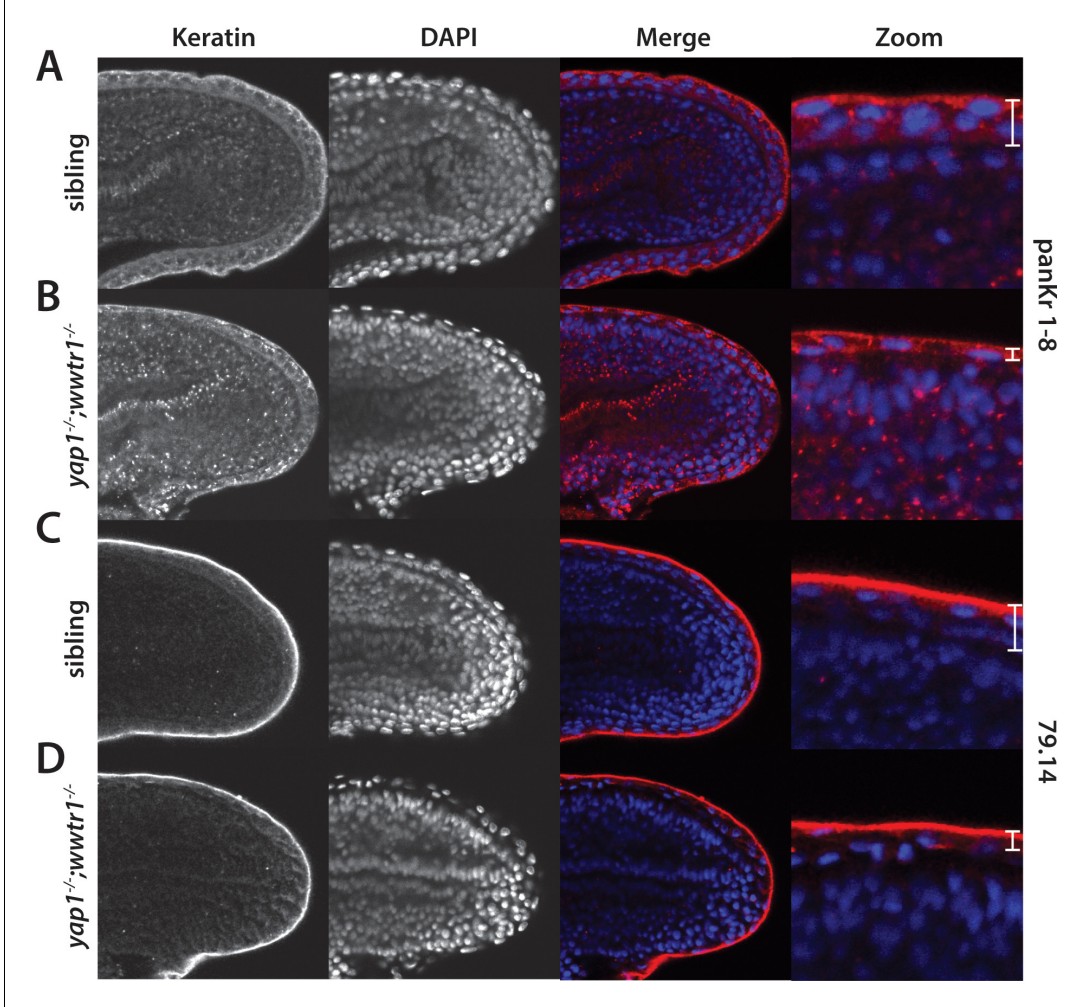

**Figure 5.** Alterations in the presumptive epidermis in *yap1;wwtr1* double mutants. Mutant and sibling embryos at the 24-somite stage were fixed, incubated with the anti-keratin antibodies panKr1-8 (**A,B**) or 79.14 (**C,D**), and then co-stained with DAPI. Confocal images of the midline of the tailbud, with anterior to the left. Note the thinner presumptive epidermis in the mutants compared to that in the siblings.
DOI: https://doi.org/10.7554/eLife.31065.015

Importantly, the increase in thickness is due to an increase in the number of Tp63-positive cells, while the number of Tp63-negative presumptive epidermal cells does not noticeably change. At the 18-somite stage, *yap1;wwtr1* double mutants appeared unaffected, with a single layer of Tp63-positive cells (*Figure 6B*, and data not shown). During the next three hours, however, while the siblings increased the thickness of the presumptive epidermis at the midline (*Figure 6C*), the double mutants were essentially unchanged from the 18-somite stage (*Figure 6D*). Examination of epidermal cell proliferation using an antibody to phospho-Histone H3 and of cell death using acridine orange did not show noticeable differences between siblings and *yap1;wwtr1* double mutants (data not shown), suggesting that the failure to increase the epidermal thickness at the midline in the double mutants is due to morphogenetic defects.

## Morphogenesis of the presumptive epidermis is altered in *yap1;wwtr1* double mutants

Previous studies of fin fold formation focusing on the dorsal side of the embryo and using cross sections of temporally staged embryos suggested that the MFF forms from shape changes of generally static cells (*Abe et al., 2007*; *Dane and Tucker, 1985*, *Figure 7A*), whereas analysis using in situ hybridization with the MFF markers *dlx5a* and *dlx6a* suggested movement of the whole presumptive

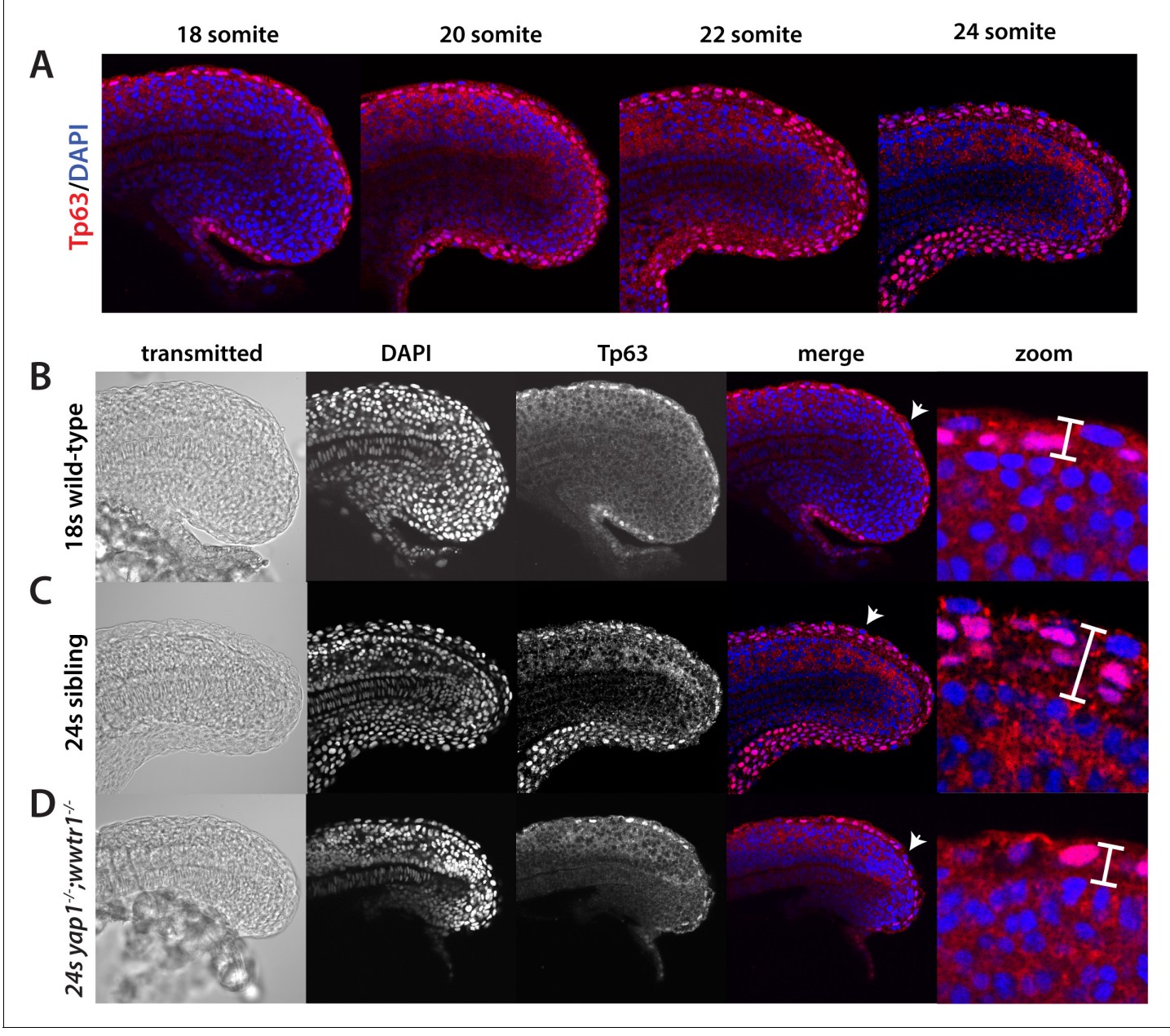

**Figure 6.** Tp63 positive cells accumulate at the nascent fin fold in sibling but not mutant embryos. Embryos were incubated with the anti-Tp63 antibody, and then co-stained with DAPI. (A) Embryos were collected at the indicated stages. Note the increase in Tp63-positive cells on the dorsal and ventral midline as the embryos age. (B–D) The Tp63-positive presumptive epidermis at the 18-somite stage (B) is one layer thick, but increases to multiple layers by the 24-somite stage in sibling (C) but not in *yap1;wwtr1* double mutant embryos (D). Confocal images of the midline of the tailbud, with anterior to the left. The zoomed in images in panels B-D are of the dorsal side, and the arrows in the merged images point to the rare presumptive epidermis Tp63-negative cells.

DOI: https://doi.org/10.7554/eLife.31065.016

epidermal sheet toward the midline as the underlying neural tissue rolled up to form the neural rod (*Heude et al., 2014*). Because previous studies had used only fixed tissue, we used live imaging to understand the cellular dynamics in both wild-type and mutant embryos as the presumptive epidermis changes shape. We also imaged the ventral fin fold as during these early stages it undergoes much more dramatic changes in overall morphology than the dorsal fin fold, although we generally observed the same results on both sides. Embryos were injected with mRNA that encodes both a membrane GFP and a nuclear version of the photoconvertible protein Kikume (NLS-Kikume/EGFP-

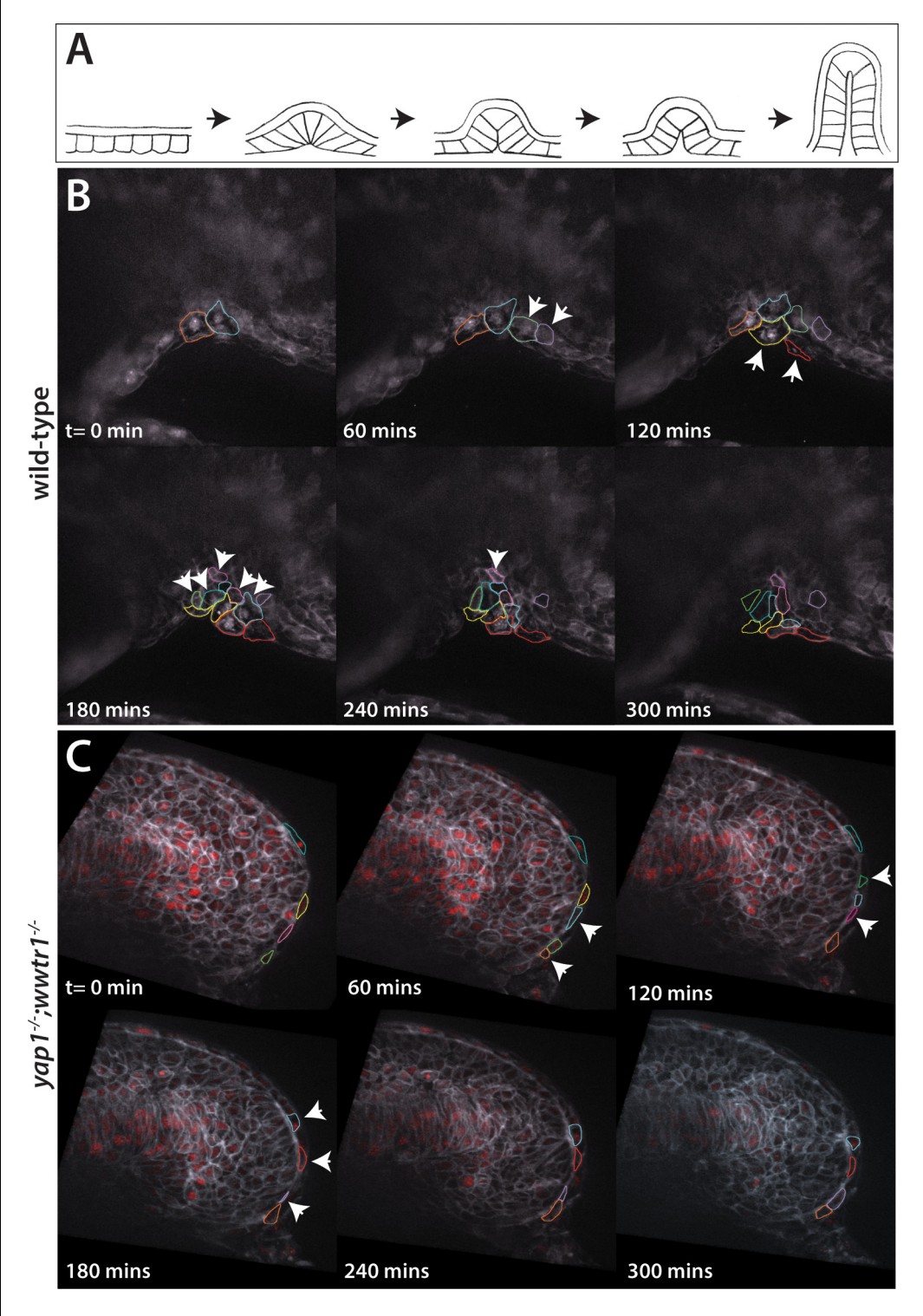

**Figure 7.** Dynamic movements of presumptive epidermal cells at the ventral fin fold. (**A**) Cartoon showing previous analysis of dorsal fin fold formation redrawn from *Dane and Tucker, 1985*. (**B**) Stills from live imaging of ventral fin fold formation in wild-type embryos taken from *Video 2*. (**C**) Stills from live imaging of ventral fin fold formation in *yap1;wwtr1* double mutants taken from *Video 3*. Images are of the midline with anterior to the left. Arrowheads point to new cells arriving at the midline. Most of the cells in the wild-type arrive basally, although two cells outlined appear at the edge of the fin fold coming from the other side of the embryo. Note that in wild-type embryos most cells that are at the midline at t = 0 min, or come to the midline at later times, stay at the midline

*Figure 7 continued on next page*

*Figure 7 continued*
whereas in the *yap1;wwtr1* double mutants cells are frequently transiently at the midline (quantified in *Figure 8*).
Lateral views of the ventral-posterior part of the embryo, with dorsal up and ventral down. Wild-type movie is a
representational movie of 21 movies examined and the mutant movie is representational of 11 movies.
DOI: https://doi.org/10.7554/eLife.31065.017

CAAX), which was photoconverted right before imaging in order to be able to follow membranes and nuclei in separate fluorescent channels, and then live imaged on a spinning disc confocal microscope between the 20- and 26-somite stages. Surprisingly, we found that in wild-type embryos the presumptive epidermal cells are highly active (*Video 2*). The cells were constantly changing shape as well as their interactions with neighbors, with new cells moving in at the basal side of the fin fold (*Figure 7B*, arrowheads). By following the newly emerging cells backwards in time through different Z-planes, we could observe cells moving in from both lateral sides toward the midline. Once cells arrived at the midline they tended to stay there (*Figure 8A*, black lines). In *yap1;wwtr1* double mutants, the cells were still moving and changing shape (*Video 3*), although not as dramatically as in wild-type. New cells showed up at the midline over time, but other cells departed such that there was no net accumulation of new cells at the midline (*Figure 7C*), in contrast to what was observed in the siblings. Following the cells over time through different Z-planes, we could watch individual cells sliding across the midline and only occasionally staying in place (*Figure 8A*, red lines). We also observed that mutant cells moved faster than wild-type cells (*Figure 8B*), and remained in an elongated shape along the body axis (length) whereas wild-type cells extended along the apical-basal axis (height, *Figure 8C*). We conclude that formation of the MFF is normally a very active process with new presumptive epidermal cells constantly moving into the midline and staying there to build up the presumptive epidermal thickening, in contrast to the bending of a presumptive epidermal sheet envisioned from the static images. In *yap1;wwtr1* double mutants, the presumptive epidermal cells are highly motile, but are unable to coalesce at the midline, resulting in the failure to form the nascent MFF.

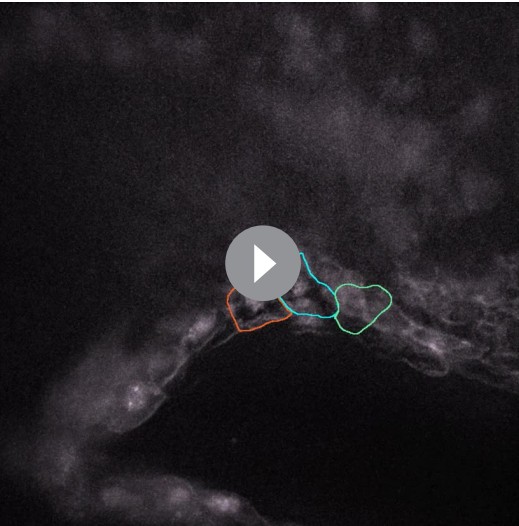

**Video 2.** A time lapse view of the presumptive epidermal cellular migration in sibling shows an accumulation of cells at the ventral midline. Cells can be seen changing contacts with neighbors, and new cells move in at the mesodermal/presumptive epidermal border. Images taken every 5 min for 5 hr starting at the 20-somite stage.
DOI: https://doi.org/10.7554/eLife.31065.019

## Yap1/Wwtr1-dependent Fibronectin assembly is required for body extension and morphogenesis of the presumptive epidermis

We wondered if there might be a common cause that connected the posterior body and presumptive epidermal morphogenetic defects, and if it was related to the observed function of Yap1/Wwtr1 in the presumptive epidermis and notochord. A series of elegant studies has demonstrated that Integrin-Fibronectin interactions play an essential role in body elongation during the somitogenesis stages as well as forming the boundaries between somites (*Dray et al., 2013*; *Jülich et al., 2005*; *Koshida et al., 2005*; *McMillen and Holley, 2015*). Intriguingly, Fn is present ubiquitously but it assembles specifically at the border between somites, at the interface between the somites and presumptive epidermis, and at the border between the somites and notochord, due to the activation of Integrins within the somitic and presomitic mesoderm at these borders (*Dray et al., 2013*; *Jülich et al., 2015*). Within the rest of the mesoderm, Cadherins act to prevent integrin activation, leading to

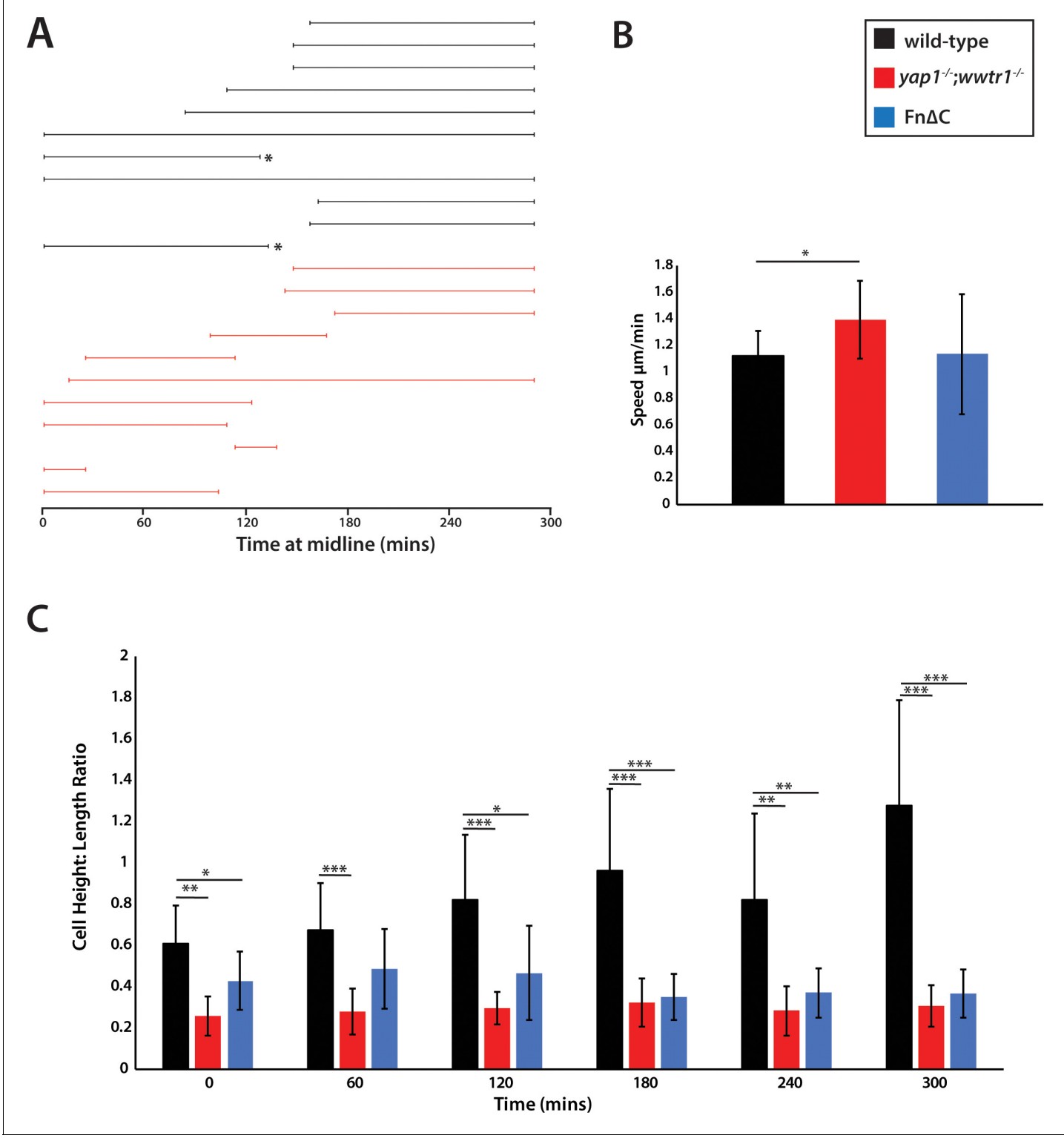

**Figure 8.** Measurements of presumptive epidermal cells at the ventral fin fold. (**A**) The time individual cells were at the midline measured in a single movie each for wild-type cells and *yap1;wwtr1* double mutant cells. Embryos were positioned on their side and filmed from the left side. When mutant cells crossed over the midline to the right side of the embryo, we stopped recording them, resulting in a truncated line. Note that while it was obvious when mutant cells left the midline because the epidermis is thin in these embryos, it was not completely clear in two of the wild-type cases (asterisks) whether cells left the midline or if they were obscured by other cells in the accumulating fin fold. (**B**) Movement speed for presumptive epidermal wild-type cells, *yap1;wwtr1* double mutant cells, and cells expressing FnΔC, all measured at the midline. (**C**) Height-Length ratio of individual presumptive

*Figure 8 continued on next page*

*Figure 8 continued*

epidermal cells at the midline, taken at various time points during the filming of an individual movie. Height is a measurement of the apical-basal distance, and length is measured as the distance along the body axis. For both panels B and C, each column in the graph represents the average of 10 cells. *p<0.05; **p<0.01; ***p<0.001.

DOI: https://doi.org/10.7554/eLife.31065.018

Fn assembly only at the borders (*Jülich et al., 2005*). Importantly, the Fn underlying the presumptive epidermis and surrounding the notochord is proposed to provide the ECM necessary for body elongation by mechanically coupling the paraxial mesoderm to the presumptive epidermis and notochord (reviewed in *McMillen and Holley, 2015*). We therefore examined Fn distribution in sibling and mutant embryos.

In sibling embryos, Fn was clearly seen in the intersomitic furrows (*Figure 9A*), and in a continuous line underlying the presumptive epidermis and surrounding the notochord at the embryonic midline (*Figure 9G*). At the lateral edges of the somites, where they contact the presumptive epidermis, rosettes of Fn fibers were observed (*Figure 9C,E*). The distribution of Fn in medaka embryos at the same stage was the same (*Figure 9—figure supplement 1*). *yap1;wwtr1* double mutants had normal Fn distribution between the somites (*Figure 9B*), whereas the Fn distribution at the lateral edges was very abnormal, with gaps in the rosettes as well as regions with enhanced Fn distribution (*Figure 9D,F*). Similarly, Fn at the midline also showed pronounced gaps in the region underlying the presumptive epidermis (*Figure 9H*, arrows). Expression of Fn around the most newly formed notochord at the posterior end of the embryo appeared mostly unaffected in *yap1; wwtr1* double mutants, but it disappeared in the region where new somites form at the anterior end of the presomitic mesoderm (*Figure 9H*, arrowheads), indicating that the notochord initially assembles Fn but is unable to maintain it. Importantly, the defects in Fn only occurred at the borders with the tissues that express Yap1/Wwtr1 (the notochord and presumptive epidermis), and not within the mesoderm where these factors are not expressed.

Embryos not expressing Fn are severely deformed, forming almost no recognizable body (*Jülich et al., 2005*). To achieve a partial knockdown of Fn function, we produced a dominant-negative Fn1b (FnΔC), since *fn1b* is the predominant *fn* expressed in the tailbuds in our RNA-seq data, based on a construct used to inhibit Fn in *Xenopus* embryos (*Rozario et al., 2009*; *Schwarzbauer and DeSimone, 2011*). Injection of 400 pg *fnΔC* mRNA into 1 cell embryos caused gastrulation defects in approximately half of the embryos. Embryos that completed gastrulation showed various degrees of body extension defects, with some embryo having phenotypes similar to those seen in *yap1;wwtr1* double mutants (*Figure 10A–C*). Embryos with an elongation defect had somites that did not become chevron-shaped (*Figure 10D,E*), and presumptive epidermal cells that did not form the multilayered MFF (*Figure 10F,G*) as in *yap1;wwtr1*

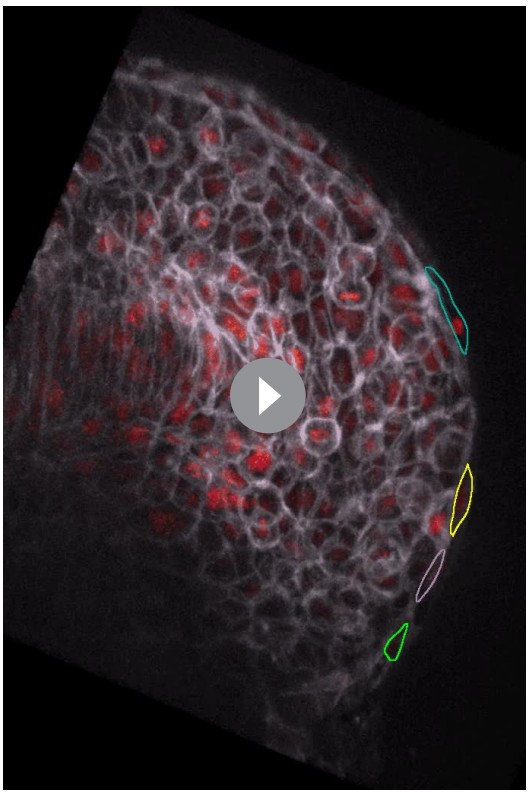

**Video 3.** A time lapse view of the presumptive epidermal cellular migration in a *yap1;wwtr1* double mutants shows that the cells are still migratory but are not accumulating at the ventral midline. Cells move into plane at the midline, but continue to migrate into other z planes instead of building up to contribute to the nascent ventral MFF. Images taken every 5 min for 5 hr starting at the 20-somite stage.
DOI: https://doi.org/10.7554/eLife.31065.020

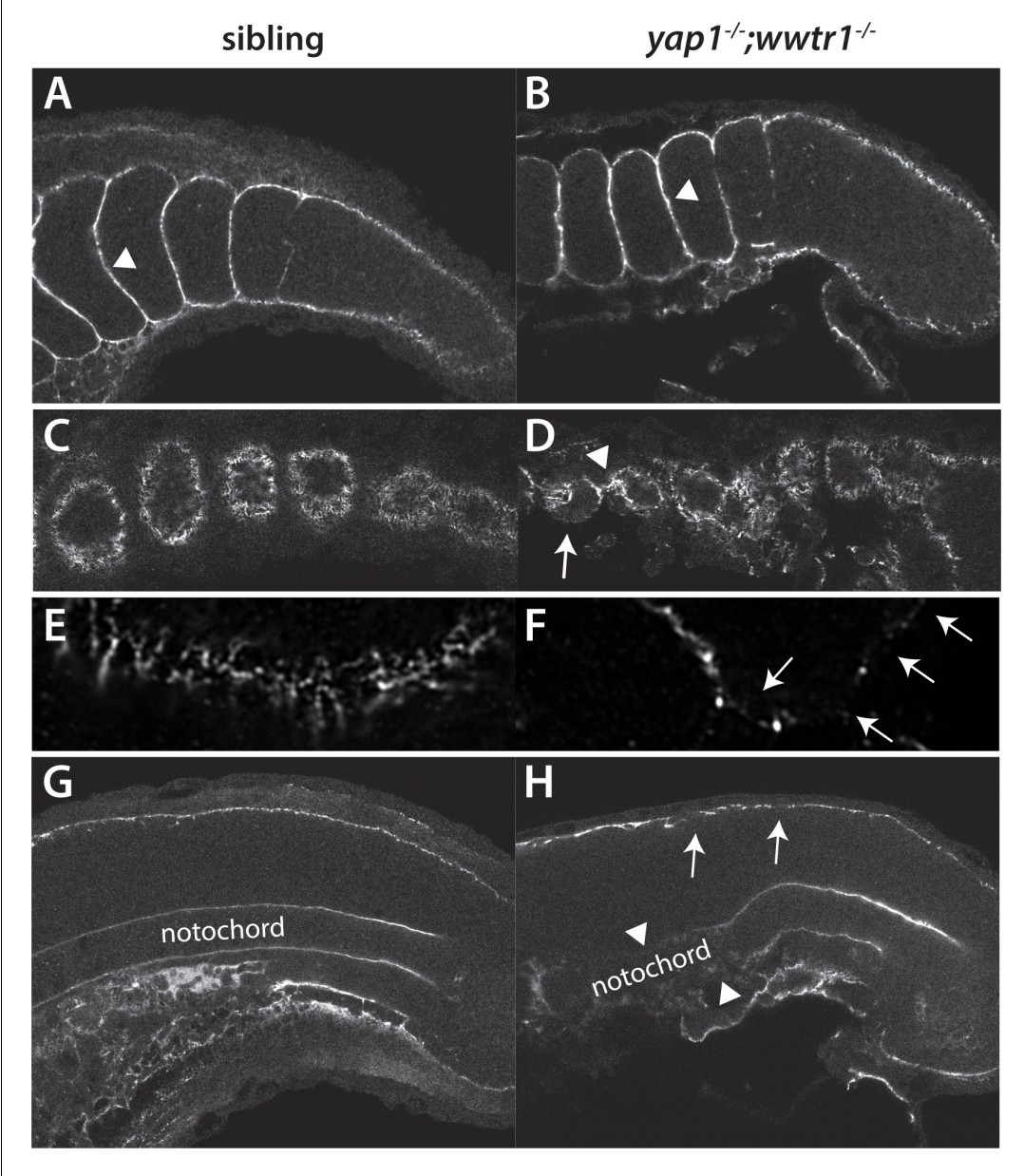

**Figure 9.** Fibronectin deposition is altered in *yap1;wwtr1* double mutants. (A,B) Intersomitic Fn deposition (arrowheads) appears unaffected in sibling and mutant embryos. Lateral confocal section. (C,D) At the lateral edges of the somites, Fn is present in rosettes where the somites contact the presumptive epidermis in sibling embryos (C), whereas in *yap1;wwtr1* double mutants there are gaps (arrow) and regions of enhanced Fn accumulation (arrowhead, (D). (E, F) Higher magnification views of the ventral region of a single somite near its lateral edge from sibling (E) and *yap1;wwtr1* double mutant (F) embryos showing altered Fn accumulation in the mutants. Arrows point to gaps in Fn deposition. (G,H) At the midline, Fn is discontinuous underneath the presumptive epidermis in *yap1;wwtr1* double mutants (arrows, (H) compared to siblings (G)). The posterior notochord Fn staining appears mostly unaffected in *yap1;wwtr1* double mutants, whereas in more anterior regions the Fn staining is absent (arrowheads), unlike in the siblings. All embryos are at 18-somites with anterior to the left.

DOI: https://doi.org/10.7554/eLife.31065.021

The following figure supplement is available for figure 9:

**Figure supplement 1.** Expression of Fn in medaka embryos.

DOI: https://doi.org/10.7554/eLife.31065.022

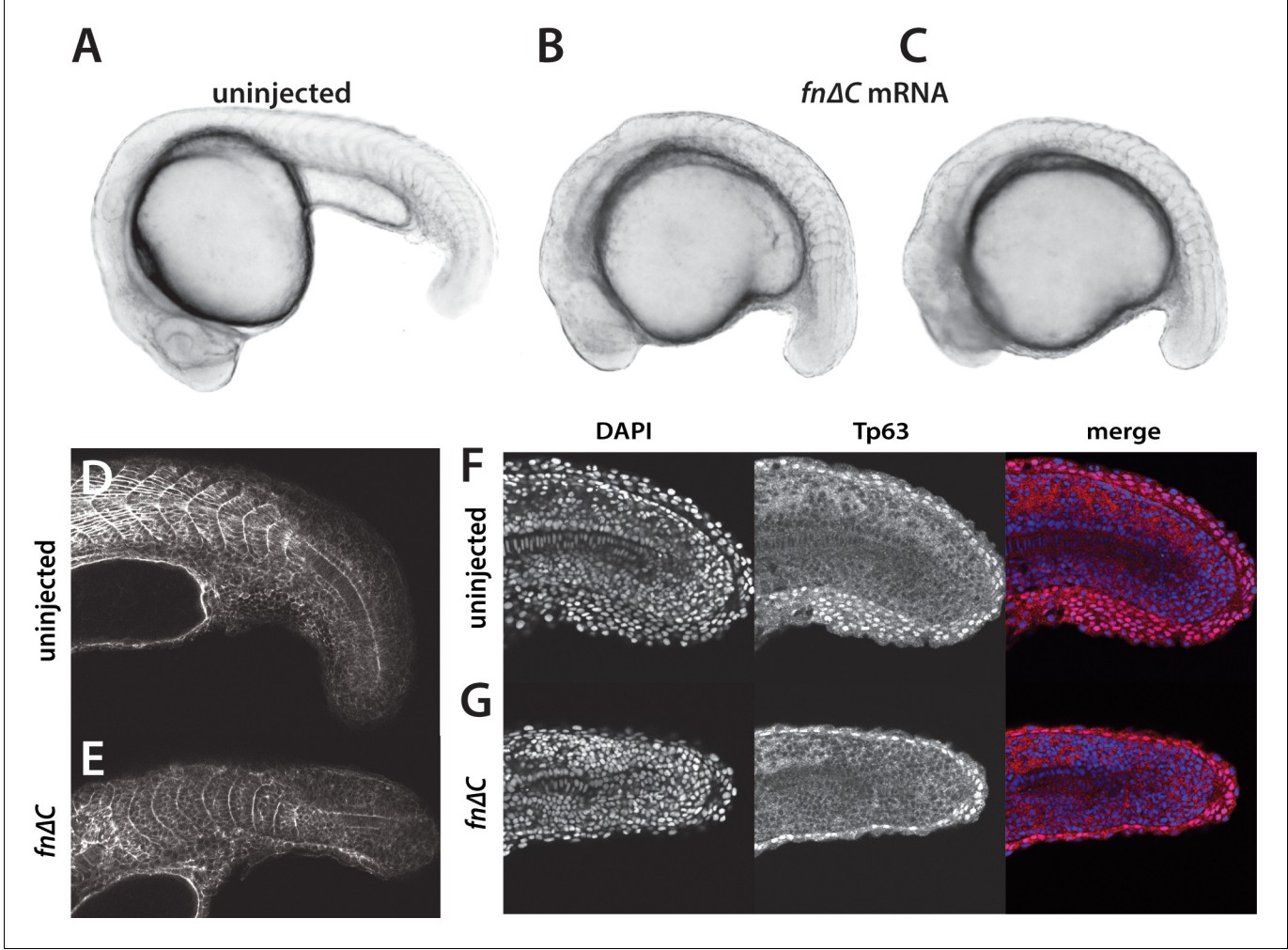

**Figure 10.** Inhibition of Fibronectin activity mimics the *yap1;wwtr1* double mutant phenotype. (**A–C**) Control embryo (**A**) and embryos injected with 400 pg *fnΔC* mRNA (**B,C**). (**D, E**) Control (**D**) and *fnΔC*-injected (**E**) embryos stained with phalloidin. Note the absence of chevron-shaped somites in panel E. (**F, G**) Control (**F**) and *fnΔC*-injected (**G**) embryos incubated with the anti-Tp63 antibody and co-stained with DAPI. Note the single layer of Tp63-positive presumptive epidermal cells in panel G (n = 12/12 embryos with a shortened tail had a single layered presumptive epidermis). All embryos are at the 24-somite stage with anterior to the left.
DOI: https://doi.org/10.7554/eLife.31065.023
The following figure supplement is available for figure 10:

**Figure supplement 1.** Movement of cells expressing FnΔC.
DOI: https://doi.org/10.7554/eLife.31065.024

double mutants. Interestingly, in these embryos somite borders still formed despite the importance of Fn for somite boundaries (*Jülich et al., 2005*; *Koshida et al., 2005*), indicating that posterior body and presumptive epidermal morphogenesis is more sensitive to disruption of Fn than is somitogenesis.

In order to understand the defects caused by disrupting Fn in more detail, we live imaged embryos expressing FnΔC (*Video 4*). In general we saw the same trends as with the *yap1;wwtr1* double mutants, although the effects were not as strong, potentially due to the variable distribution of protein from injected mRNA. As was observed in the fixed tissue (*Figure 10G*), the presumptive epidermis in the FnΔC-expressing embryos was a single layer thick (*Figure 10—figure supplement 1A*). As with the *yap1;wwtr1* double mutant cells, the cells expressing FnΔC often remained at the midline for short intervals of time (*Figure 10—figure supplement 1B*). The FnΔC-expressing cells showed variable rates of movement, with some cells moving faster than wild-type cells and none

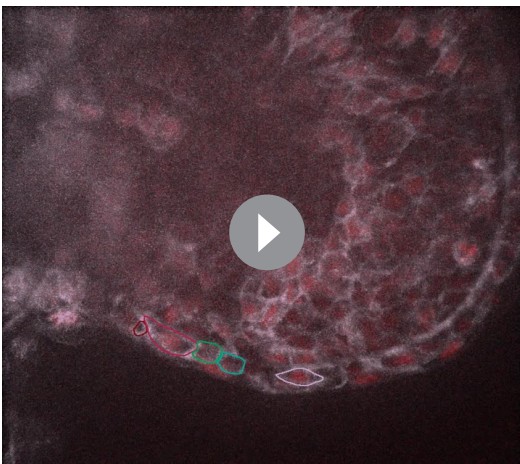

**Video 4.** A time lapse view of the presumptive epidermal cellular migration in an embryo injected with 400 pg *fnΔC* mRNA. As with the *yap1;wwtr1* double mutants, the presumptive epidermal cells are still migratory but are not accumulating at the ventral midline. Images taken every 5 min for 5 hr starting at the 20-somite stage.
DOI: https://doi.org/10.7554/eLife.31065.025

moving slower (*Figure 8B*). Moreover, FnΔC-expressing cells did not adopt the wild-type morphology over time, and instead remained in an elongated shape along the body axis as was observed in the *yap1;wwtr1* double mutant cells (*Figure 8C*). Thus, disruption of Fn causes similar effects to those see in *yap1;wwtr1* double mutants, both in terms of overall phenotype as well as in the behavior of presumptive epidermal cells.

Since Yap1/Wwtr1 are mechanosensors and regulated by extracellular signals (*Piccolo et al., 2014*), we asked whether alterations in Fn might also regulate their activity. FnΔC expressing embryos exhibiting a *yap1;wwtr1* double mutant-like phenotype (*Figure 10B,C*) were analyzed using the anti-Yap1 antibody. Expression of Yap1 was the same as in wild-type (*Figure 2A*), with strong nuclear expression in the epidermis and notochord (data not shown). Thus, the alterations in Fn seen in the *yap1;wwtr1* double mutants are not in turn regulating Yap1.

## Reduced adhesion in *yap1;wwtr1* double mutants

The disruptions in Fn surrounding the notochord and under the presumptive epidermis suggested that these tissues might have reduced adhesion to their neighboring tissues. An undulating notochord is indicative of reduced mechanical coupling between the notochord and paraxial mesoderm (*Dray et al., 2013*), and we also observed an undulating notochord in *yap1;wwtr1* double mutant embryos (*Figures 1F* and *9H*). We also observed that the presumptive epidermis seemed less firmly attached to the underlying tissue when we microdissected tailbuds for RNA-seq analysis. To examine this phenotype more directly, we injected embryos with mRNA encoding a membrane-targeted GFP and then live imaged embryos at the 16- and 20-somite stages. Whereas the presumptive epidermis remained attached to the underlying tissues in sibling embryos (*Figure 11A,C*), in *yap1;wwtr1* double mutant embryos the presumptive epidermis began to separate from the somites at the 16-somite stage, and this phenotype was very pronounced at the 20-somite stage (*Figure 11B,D*). Taken together, our results support a model in which Yap1 and Wwtr1 regulate gene expression in the notochord and presumptive epidermis in order to promote cell adhesion and fibronectin deposition, which is necessary for both posterior body elongation and presumptive epidermal morphogenesis. We elaborate on this model below.

## Discussion

We show here that zebrafish *yap1;wwtr1* double mutants have a severe defect in body elongation beginning at approximately the 15–16 somite stage. Although this process was not studied in medaka, the *yap1/hirame* mutants also lack an extended tail (Figure 1 in *Porazinski et al., 2015*), demonstrating that this is a conserved process. We also found that *yap1;wwtr1* double mutants fail to form the nascent MFF, which was not mentioned for *yap1/hirame*, most likely because the flattening of the medaka embryo precluded conclusions about the state of the MFF. What we did not find similar was the flattening of the body in zebrafish *yap1;wwtr1* double mutants (*Figure 1B,E,F*), which was the basis for the proposed key role of Yap in opposing the forces of gravity (*Porazinski et al., 2015*). Why these two species differ in this regard is not clear at this point but possibly due to the fact that medaka has a hard chorion whereas zebrafish has a very soft chorion, and thus medaka may have evolved to specifically resist the pressure of the embryo against the chorion.

We find that Yap1 and Wwtr1 are not present ubiquitously throughout the embryo but instead are both specifically localized to the presumptive epidermis and notochord. Intriguingly, we also find

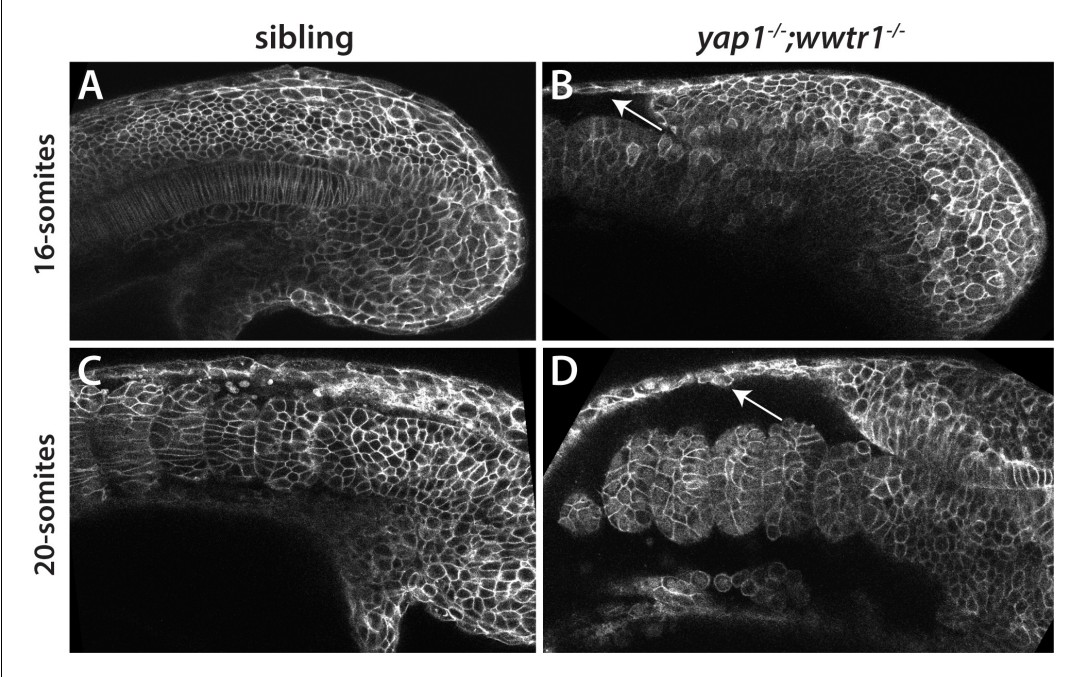

**Figure 11.** Reduced adhesion of presumptive epidermis in *yap1;wwtr1* double mutants. (A–D) Embryos were injected with mRNA encoding a membrane-localized form of GFP and imaged at the indicated stages. (A,C) Sibling embryos. (B,D) *yap1;wwtr1* double mutant embryos. Note the progressive separation of the epidermis from the somites (arrows), which increases from the 16-somite to the 20-somite stage. Four of four mutant embryos at the 16-somite stage exhibited the tissue separation phenotype and three of three at the 20-somite stage.

DOI: https://doi.org/10.7554/eLife.31065.026

that medaka Yap1 is similarly localized, which was not previously known. The localization of the zebrafish proteins is in strong accord with our RNA-seq data, which shows major changes in gene expression in presumptive epidermal and notochordal genes. While these two tissues are very different in function and lineage, we see alterations in Fn assembly, which underlies the presumptive epidermis and overlies the notochord. Importantly, we do not see defects in Fn assembly between the somites, which supports the finding that Yap1 and Wwtr1 do not function within the mesoderm, and also shows that there is not a general defect in Fn assembly. Defects in Fn assembly were also seen in *yap1/hirame* mutant embryos between the lens and retina (*Porazinski et al., 2015*). Surprisingly, altered Fn in *yap1/hirame* mutants was also reported throughout the neural tube whereas we do not find any Fn assembled in the neural tube in zebrafish, as previously reported (*Dray et al., 2013*), or in medaka, which is more in keeping with the normal assembly of extracellular matrix between tissues. The importance of Fn in zebrafish is shown with the use of a dominant-negative mutant, FnΔC, that phenocopies both the extension defect and presumptive epidermal cell migration defects seen in *yap1;wwtr1* double mutants. These results, taken together with our evidence of decreased adhesion of the notochord and presumptive epidermis to the surrounding tissues, supports a model in which Yap1 and Wwtr1, functioning redundantly within the presumptive epidermis and notochord, work to promote posterior body elongation and presumptive epidermal morphogenesis necessary for MFF formation (*Figure 12A*).

## Regulation of posterior extension by Yap1 and Wwtr1

The defects in *yap1;wwtr1* double mutants begin at the 15–16 somite stage, which is the time when the tailbud everts from the body, and the body begins to achieve a straight conformation along the anterior-posterior axis rather than being curved around the yolk. Indeed, *yap1;wwtr1* double mutants remain in a curved conformation (*Figure 1B*). Part of this defect can be attributed to defects in the notochord as we found that *noto* mutants also begin to show a phenotype at the 15–16 somite stage, but the defect is much less severe than that seen in *yap1;wwtr1* double mutants. These

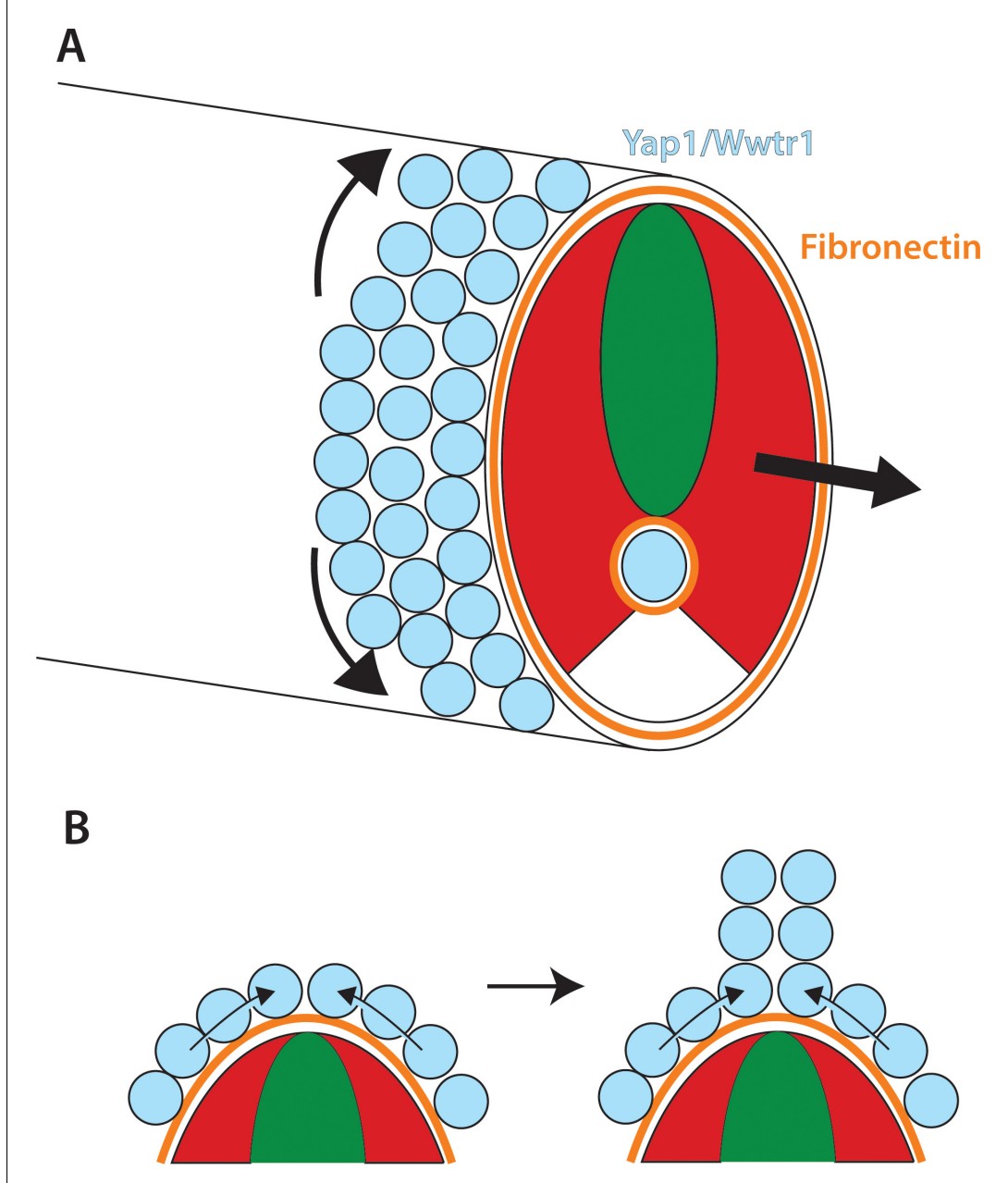

**Figure 12.** Role of Yap1/Wwtr1 in somite stage morphogenesis. (**A**) Yap1 and Wwtr1 are expressed specifically in the presumptive epidermis and notochord (light blue), where they promote cell adhesion and Fn assembly (orange). These processes are essential for both the movement of presumptive epidermal cells dorsally and ventrally to form the MFF, and for force transmission of the underlying tissues to allow the elongation of the posterior body. (**B**) At the midline, cells are converging from either side and push against each other to form the dorsal (shown) and ventral MFFs. We propose that adhesion to Fn is required for cells to be able to create the necessary force to allow them to generate the MFF. The presomitic mesoderm is shown in red and the neural tube is green.

DOI: https://doi.org/10.7554/eLife.31065.027

results point to the presumptive epidermis, the other major site of Yap1/Wwtr1 expression, as also being involved in formation of the posterior body. The key factor linking the presumptive epidermis and notochord to body extension is the ECM, and specifically Fibronectin (reviewed in *McMillen and Holley, 2015*). Fn couples the underlying mesoderm to the notochord and presumptive epidermis, which allows forces from convergence-extension within the paraxial mesoderm to allow the body to extend. The notochord also undergoes convergence-extension, and this helps the

body extend as long as the notochord is coupled to the mesoderm (*Glickman et al., 2003*; *Keller et al., 1992*; *Steventon et al., 2016*; *Weliky et al., 1991*; *Wilson and Keller, 1991*). Finally, the spinal cord, which is connected to the presumptive epidermis, and the notochord undergo volumetric growth as the body extends, further assisting in the extension of the body (*Steventon et al., 2016*). In *noto* mutants, the notochord component is removed but the presumptive epidermis remains. When this process is disrupted such as in the case of our dominant-negative Fn injections, or when Integrin function is knocked down (*Dray et al., 2013*), posterior extension is blocked. However, unlike in the Integrin knockdown experiments, *yap1;wwtr1* double mutants form normal somites because the intersomitic Fn deposition is unaffected. Intriguingly, the observation that the dose of dominant-negative Fn we used produced embryos with defective body extension but normal somites argues that body extension is more sensitive to disruption than the formation of somite boundaries. Possibly because the Fn underlying the presumptive epidermis and surrounding the notochord is under much more force from the convergence-extension movements of the presomitic mesoderm that extend the body (*Steventon et al., 2016*) than is the Fn that separates the somites, this pool of Fn may be much more sensitive to even minor perturbations.

We also found decreases in notochordal and presumptive epidermal cell adhesion in *yap1;wwtr1* double mutants. In living embryos we see a separation of the prospective epidermis from the underlying somitic tissue, which becomes progressively more pronounced as the overall phenotype of the embryos increases. In addition, the undulating notochord we observed in the mutants is indicative of decreased mechanical coupling between the notochord and paraxial mesoderm as is also seen in Integrin knockdown embryos (*Dray et al., 2013*). Since we see no changes in *fibronectin* expression levels and little or no changes in *integrin* levels in our RNA-seq analysis (*Supplementary file 1*), our results suggest that there are defects within the presumptive epidermis and notochord that results in a failure to assemble Fn. While recent results have shown how Cadherins suppress Fibronectin assembly between adjacent cells within the paraxial mesoderm by keeping Integrins in an inactive conformation (*Jülich et al., 2015*), the factors that regulate the activation of Integrin within the early embryo are not known. Our RNA-seq data of genes down-regulated in *yap1;wwtr1* double mutants provides a valuable dataset of genes worth further investigation, most of which have not been analyzed even at the level of in situ hybridization let alone functionally studied.

## Presumptive epidermis morphogenesis in *yap1;wwtr1* double mutants

Our analysis of the presumptive epidermis in *yap1;wwtr1* double mutants has provided new insight into the mechanisms that form the nascent MFF from presumptive epidermal cells. We find that the presumptive epidermal cells are highly active and constantly changing shape when imaged in live embryos. They actively move toward the dorsal and ventral midlines as individual cells and then change their overall shape, with an increase along the apical-basal axis. At later stages, as laminin is used to stabilize the nascent MFF (*Webb et al., 2007*) by forming connections between apposing cells (*Dane and Tucker, 1985*), we expect that cell motility is reduced allowing the unpaired fins to extend away from the body. In *yap1;wwtr1* double mutants, the presumptive epidermal cells are still highly motile but they transit through the midline rather than staying there, and they remain in an elongated shape along the body axis with no increase in the apical-basal direction, all resulting in a failure to form the MFF. Intriguingly, we can largely recapitulate this effect by inhibiting Fn assembly using dominant-negative Fn injections. In these embryos, the body fails to elongate, the presumptive fin fold does not form, and the cells transit through the midline and remain in an elongated shape. Interestingly, inhibition of cell movement with the myosin inhibitor Blebbistatin did not result in defects in Fn assembly under the epidermis (or around the notochord), demonstrating that the Yap1/Wwtr1-dependent defects in Fn assembly cause migration defects, rather than defective migration of cells leading indirectly to altered Fn assembly (data not shown). These results suggest that in addition to its role in posterior elongation, the Fn matrix provides a substrate for the presumptive epidermal cells to adhere to, which allows them to converge from the right and left sides of the embryo and push against each other to initiate MFF formation (*Figure 12B*). We call this process concerted migration, since the fin fold forms from the action of the presumptive epidermal cells migrating dorsally or ventrally and pushing against each other, creating the force to form the fin fold, and increasing the cells' height. These processes then allow the basal surfaces of juxtaposed MFF cells to adhere to each other, with the resulting formation of cross-linking fibers to stabilize the

structure (*Dane and Tucker, 1985*). Thus, Fn does double duty in extending the body and creating the fins that the elongated body uses to swim.

## The Yap1/Wwtr1 transcriptional program

Transcriptional analysis of the genes regulated in the tailbud by Yap1/Wwtr1 identified many genes with large changes in expression, including 8 of our top 12 hits which had not previously been analyzed by in situ hybridization in the early embryo. Importantly, all these genes were expressed in the presumptive epidermis or notochord, exactly the tissues in which we found Yap1 and Wwtr1 expressed. While many of the genes further down the list have not been characterized, those that have are also expressed in the notochord or presumptive epidermis. An outlier at position 36 on our list with a 4.4-fold change is myosin light chain (*mylpfa*). However, in the tailbud *mylpfa* is expressed specifically in the adaxial cells (*von Hofsten et al., 2008*), the small population of future slow muscle cells that originate directly next to the notochord. Since adaxial cell formation depends on interactions with the notochord (*Jackson and Ingham, 2013*), the decreased *mylpfa* expression may be a secondary effect of alterations to the notochord.

Our results suggest that the *arhgaps* are unlikely to be the key genes regulated by Yap1/Wwtr1 in zebrafish. Two weakly expressed presumptive epidermal *arhgaps*, *arhgap27* and *arhgap12a*, are found far down on our list of dysregulated genes ordered by fold-change (positions 254 and 300, respectively), among a long list of genes that change approximately 2-fold. Moreover, the only notochord expressed *arhgap*, *arhgap17a*, is not changed in its expression levels in *yap1;wwtr1* double mutant tails. In human 3D spheroids formed using a human retina pigmented epithelial cell line, *ARHGAP18* was identified as one of 40 genes down-regulated when YAP levels were reduced, and reduction of *ARHGAP18* by siRNA caused a similar phenotype to siRNA reduction of YAP (*Porazinski et al., 2015*). In medaka *yap1* mutants, however, *arhgap18* was reduced just 24% from control levels, and inactivation of *arhgap18* did not produce a recognizable phenotype (*Porazinski et al., 2015*). In zebrafish, *arhgap18* is not expressed in the tailbud, nor is its closest relative, *arhgap28* (37% amino acid identity). Similarly, *ARHGAP29* was found to be down-regulated 50% in a human gastric cancer cell line after YAP1 knock out, but the levels of *arhgap29* were unchanged in the tailbud of zebrafish *yap1;wwtr1* double mutants. These results show that the Yap1 transcriptional program is context dependent and the results found in cell culture can not be inferred to be true in the embryo (*Qiao et al., 2017*).

Our data also show that Yap1/Wwtr1 is mostly regulating a completely different suite of genes in the presumptive epidermis than in the notochord. Of the known genes expressed in both tissues, the highest scoring gene is *laminin α5* (position 86), which is decreased 3-fold in *yap1;wwtr1* double mutants. However, we previously identified a mutant in this gene and saw no effect on posterior body formation, and no effect on fin-fold formation prior to 30 hpf (*Webb et al., 2007*). Our results suggest therefore that Yap1 and Wwtr1 are responsible for broad transcriptional regulatory networks within both the presumptive epidermis and notochord that allow these cells to adhere to the surrounding tissue and assemble the extracellular matrix.

Our results also reveal that Yap1/Wwtr1 do not activate a generic transcriptional program in the early embryo, and instead work in concert with tissue specific factors. This was clearly shown in the case of *ecrg4b* which is expressed in the epidermis but not in the notochord, whereas Yap1 and Wwtr1 are active in both tissues. Moreover, when a constitutively active Yap was ubiquitously expressed, *ecrg4b* expression increased only in the epidermis but was not activated in other tissues. Finally, well known Hippo pathway transcriptional targets such as *cyclin D1* and *aurora kinase A* were not down-regulated in the *yap1;wwtr1* double mutants according to our RNA-seq data, also supporting the idea that there is not a generic pathway activated by these factors. Instead, our data argue that Yap1 and Wwtr1 enhance the expression of particular genes in concert with tissue-specific transcription factors, but they themselves do not provide tissue specificity.

Intriguingly, a newly published paper has shown that YAP1, which is expressed in the murine presomitic mesoderm (PSM), has an essential role in repressing oscillations of the somite clock (*Hubaud et al., 2017*). Since this repression is a key part of the regulatory system (*Hubaud et al., 2017*), and since Yap1 (or Wwtr1) is not expressed in the fish PSM in either zebrafish or medaka, it raises the question of what factor might be playing a role in fish similar to that identified for YAP1 in mouse. As the key targets of YAP1 in the murine PSM are identified, it will be important to compare their regulation in fish.

# Materials and methods

## Key resources table

| Reagent type (species) or resource | Designation | Source or reference | Identifiers | Additional information |
|---|---|---|---|---|
| Gene (*D. rerio*) | *yap1; wwtr1* | NA | ZFIN_ID:ZDB-GENE-030131–9710; ZFIN_ID :ZDB-GENE-051101–1 | |
| Strain, strain background () | | | | |
| Genetic reagent (*D. rerio*) | *yap1*[bns19]; *wwtr1*[bns35] | this paper | | CRISPR-mediated mutation. 41 bp deletion in exon 1. p.Ile39Argfs*72; CRISPR-mediated mutation. 29 bp insertion in exon 2. p.Pro145Glnfs*15 |
| Genetic reagent (*D. rerio*) | *Tg(hsp70:RFP-DNyap) zf621* | Source: Poss lab; Reference PMID: 26209644 | | |
| Genetic reagent (*D. rerio*) | *Tg(hsp70:RFP-CAyap) zf622* | Source: Poss lab; Reference PMID: 26209644 | | |
| Genetic reagent (*D. rerio*) | *noto n1* | Source ZIRC; Reference PMID: 7477317 | | |
| Biological sample (*D. rerio*) | NA | NA | | Whole tail from the tailbud to the third newest somite (S-III). 16–18 somite-stage embryos. Tissue samples for RNA-seq and real-time qPCR. |
| Antibody | anti-WWTR1 (rabbit monoclonal) | Cell Signaling | D24E4 | *Figure 2—figure supplement 2A* 1:200; 1:100 *Figure 2—figure supplement 2B,C* |
| Antibody | anti-Yap1 (rabbit polyclonal) | Cell Signaling | 4912 | Dilution 1:100 |
| Antibody | anti-Tp63 (rabbit polyclonal) | GeneTex | GTX124660 | Dilution 1:800 |
| Antibody | anti-Fibronectin (rabbit polyclonal) | Sigma | F3648 | Dilution 1:100 |
| Antibody | pankr1-8 (mouse monoclonal) | Progen Biotechnik | 61006 | Dilution 1:10 |
| Antibody | anti-keratin (mouse monoclonal) | Developmental Studies Hybridoma Bank | 79.14 | Dilution 1:10 |
| Recombinant DNA reagent | dominant-negative Fn in CS2 expression vector | this paper | | A fragment of zebrafish Fn1b coding region from aa 1–630 was inserted into the CS2 vector as described in the Methods. |
| Recombinant DNA reagent | NLS-Kikume/ EGFP-CAAX in the CS2 expresion vector | this paper | | |

## Zebrafish lines and embryos

The *yap1*[bns19] and *wwtr1*[bns35] CRISPR mutant lines are described elsewhere and genotyped as described (*Lai et al., 2017*). Double mutant *yap1*[-/-];*wwtr1*[-/-] embryos were identified by phenotype whereas the sibling embryos had a wild-type phenotype. The *noto*[n1] line (*Talbot et al., 1995*) was obtained from the Zebrafish International Resource Center and genotyped using the provided instructions. The transgenic lines *Tg(hsp70:RFP-DNyap)*[zf621] and *Tg(hsp70:RFP-CAyap)*[zf622] (*Mateus et al., 2015*) were obtained from Dr. Ken Poss (Duke). These transgenic fish were genotyped by PCR with the primers Yap 394 F (ACTACACCTGCGACTTCAAGA) and Yap 394 R

(CCAGCAGTACCTGCATCTGTA), which produce a 394 bp fragment. Embryos were heat shocked at the 4-somite stage at 40 C for 30 min in a circulating water bath. Fixed medaka embryos were kindly provided by Siddharth Ramakrishnan (University of Puget Sound).

## Analysis of cell movement

A construct was produced in the vector CS2 that has a nuclear localized version of the photoconvertible protein Kikume, followed by the viral 2A sequence, followed by a membrane localized EGFP (NLS-Kikume/EGFP-CAAX). This was used to make mRNA using the mMessage Machine SP6 kit (Ambion). 15 pg of mRNA were injected at the 1 cell stage, and imaged on a spinning disc confocal microscope (3I) using a 40x water immersion lens. Embryos were mounted laterally in a glass-bottomed dish and immobilized with tricaine. Images were acquired every 5 min with Z-axis intervals of 1 μm for 5 hr. Image processing was done using the Fiji plugin for ImageJ.

## Inhibition of Fn

A fragment of the Fn1b coding region from the start codon to amino acid 630 was amplified with the primers ggcggcggatccgAAATGACCCGTGAGTCAGTAAAGAG and ggcggcatcgATGGGAA TTGGGCTGATTTCCAG, and cloned into the *BamH1* and *ClaI* sites of the CS2 vector to make mRNA using the mMessage Machine SP6 kit. 1 cell embryos were injected with 100 or 150 pg of mRNA.

## In situ hybridization, immunofluorescence and Phalloidin Staining

Alkaline phosphatase in situ hybridizations were performed using digoxygenin-labeled probes. The primers used to produce probes for this paper not previously published are shown in *Supplementary file 3*, and all were cloned into the Bluescript vector (Stratagene). Fluorescent in situ hybridizations were as described (*Lauter et al., 2011*), except that a primary antibody (anti-Yap or anti-Wwtr1) was used after the first tyramide step, followed by the use of a fluorescent secondary antibody. For immunofluorescence, embryos were fixed in 4% paraformaldehyde (PFA) 2 hr at room temperature or overnight at 4 C, except for the keratin antibodies, which were fixed with Dent's fixative for 3 hr at room temperature. The anti-Tp63 antibody was purchased from GeneTex (Irvine, CA, USA; GTX124660) and used at a 1:800 dilution according to the manufacturer's protocol. The anti-Fibronectin antibody was purchased from Sigma (F3648) and used at a 1:100 dilution following a protocol kindly provided by Scott Holley. The anti-Yap (4912) and Wwtr1 (D24E4) antibodies were purchased from Cell Signaling Technology and both were used at a 1:100 dilution with standard immunofluorescence conditions. Note that the Wwtr1 antibody is described as a Yap/Taz antibody, but in zebrafish it is Wwtr1-specific (*Figure 2—figure supplement 2*; *Lai et al., 2017*; *Miesfeld et al., 2015*). The keratin antibodies, pankr1-8 (Progen Biotechnik, Heidelberg, Germany; 61006) and 79.14 (Developmental Studies Hybridoma Bank) were both used at a 1:10 dilution with standard immunofluorescence conditions. For phalloidin staining, embryos were fixed in 4% PFA, digested for 10 mins with 10 ug/ml proteinase K, then re-fixed in PFA and incubated in Alex Fluro 568 phalloidin (Invitrogen) overnight at 4° C. All whole mount in situ images were performed on a minimum of 12 embryos from replicate experiments and confocal images were from a minimum of 10 samples from replicate experiments.

## RNA-seq analysis

Two replicate samples of twenty-four tails of 16–18 somite stage embryos from a cross of $yap1^{+/-}$; $wwtr1^{+/-}$ adults were collected on seven separate occasions. Embryos from a cross of $yap1^{+/-}$; $wwtr1^{+/-}$ adults were separated into mutants or siblings solely by phenotype. We collected the tails in a manner that randomized the collection order of the samples (mutant, siblings) and replicates (replicate 1, 2 and 3; a third one as backup). The collection order was not identical from one collection to the next. Tails are tissues of the whole tail from the tailbud to the third newest somite (S-III). RNA from two replicates of mutant and sibling samples were isolated using the miRNeasy micro Kit (Qiagen) and treated with in-column DNase digestion. Quality checks on RNA for library preparation were performed with a BioAnalyzer 2100 (Agilent Technologies) and LabChip Gx Touch 24 (Perkin-Elmer). A total of 300 ng of total RNA was used as input for Truseq Stranded mRNA library preparation. Sequencing was performed on the NextSeq 500 instrument (Illumina) using v1 chemistry,

obtaining a minimum of 10–15M reads (75 bp paired-end) per library. Quality checks were performed on raw reads with FastQC (http://www.bioinformatics.babraham.ac.uk/projects/fastqc). Adapter sequences and poor quality base calls were trimmed from the raw reads with Trimmomatic v0.33 (parameters: LEADING:3 TRAILING:3 SLIDINGWINDOW:5:20 CROP:500 MINLEN:15) (Bolger et al., 2014). The processed reads were then aligned to Zv10 version of the zebrafish genome with STAR 2.4.0a (parameters: –outFilterScoreMinOverLread 0 –outFilterMatchNminOverLread 0 –outFilterMatchNmin 30 –outFilterMultimapNmax 999 –alignEndsProtrude 10 ConcordantPair) (Dobin et al., 2013). The resulting bam alignment file was passed through the Cufflinks suite v2.2.1 with standard parameters (Trapnell et al., 2012) to calculate the Fragments Per Kilobase of transcript per Million mapped reads (FPKM) for each gene as well as the statistics on the fold-changes. The summarized data can be found in Supplementary file 1 and the complete dataset is deposited on GEO with the accession number GSE102606.

### Gene set enrichment analysis (GSEA)

Enrichment of genes for Gene Ontology (GO) terms and Kegg Pathways were performed using the Database for Annotation, Variation and Integrated Discovery (DAVID) web tool (Huang et al., 2009). Genes that are significantly downregulated (adjusted p-value<0.05, and sibling FPKM >8500) were defined as 'Gene List' in the input, and compared against all zebrafish genes. GO terms and Kegg Pathways were significantly enriched when the corrected $P$-value (Benjamini) <0.05.

### Previously identified Yap1/Wwtr1 target genes

We identified Yap1/Wwtr1 target genes from published datasets (Zhang et al., 2009; Zhao et al., 2008). The list of genes induced by YAP1 overexpression in MCF10A cells are found in Supplementary file 3 of Zhang et al., 2009 and Supplementary file 3 of Zhao et al. (2008). These two lists were overlapped to obtain a list of Yap1/Wwtr1 target genes. We then identified which of these genes are significantly downregulated in the tail buds of yap1;wwtr1 double mutants.

### Real-time PCR (qPCR)

Tail bud samples were collected in the same manner as in the RNA-seq experiment. A total of 4 biological replicates (pool of 6 tail buds) from siblings and mutants were collected. Total RNA was isolated with the Direct-zol kit (Zymogen, R2061) following manufacturer's instructions. RNA-cDNA reaction was performed with the Maxima First Strand cDNA Synthesis Kit for RT-qPCR (Thermo Scientific, K1671). A total of 80–130 ng of RNA per sample was used in the cDNA synthesis reaction. Primers for the qPCR reactions can be found in Supplementary file 4. The DyNAmo Flash SYBR Green qPCR Kit (Thermo Scientific, F415) was used for the qPCR reaction. The amplification cycle was: 95℃ for 15 s; 65℃ for 15 s; 72℃ for 5 s. All genes of interest Cq values were normalized to $rpl13$ Cq values ($\Delta$Cq). $\Delta$Cq$_{sibling}$ – $\Delta$Cq$_{mutant}$ is the log$_2$(Fold-change) for each gene of interest. The one-tailed two-sample t-test was performed to determine whether the $\Delta$Cq values are significantly different between siblings and mutants. $P$-values were corrected for multiple hypotheses testing with the FDR (false-discovery rate) method.

## Acknowledgements

We thank Young Kwon for suggesting (and providing) the Yap1 antibody, Scott Holley for sharing his Fibronectin staining protocol, Siddharth Ramakrishnan for providing medaka embryos, as well as Michelle Collins and Stefan Günther for help preparing the RNA-seq samples.

## Additional information

#### Competing interests

Didier YR Stainer: Senior editor, eLife. The other authors declare that no competing interests exist.

## Funding

| Funder | Grant reference number | Author |
|---|---|---|
| National Institutes of Health | GM079203 | David Kimelman<br>Natalie L Smith |
| Max-Planck-Gesellschaft | | Jason Kuan Han Lai<br>Didier YR Stainier |

The funders had no role in study design, data collection and interpretation, or the decision to submit the work for publication.

## Author contributions

David Kimelman, Conceptualization, Supervision, Funding acquisition, Writing—original draft, Project administration, Writing—review and editing, Experiments; Natalie L Smith, Investigation, Writing—review and editing, Experiments; Jason Kuan Han Lai, Data curation, Writing—review and editing, Experiments; Didier YR Stainier, Supervision, Funding acquisition, Investigation, Project administration, Writing—review and editing

## Author ORCIDs

David Kimelman [ID] http://orcid.org/0000-0002-9261-4506
Didier YR Stainier [ID] http://orcid.org/0000-0002-0382-0026

## Ethics

Animal experimentation: This study was performed in strict accordance with institutional (UW and MPG) and national ethical and animal welfare guidelines. All of the animals were handled according to approved institutional animal care protocols (Permission No. B2/1068 for DS and IACUC protocol 2387-02 for DK).

## Decision letter and Author response

Decision letter https://doi.org/10.7554/eLife.31065.036
Author response https://doi.org/10.7554/eLife.31065.037

# Additional files

## Supplementary files

• Supplementary file 1. RNA-seq of tailbuds Spreadsheet showing gene changes between siblings and *yap1;wwtr1* double mutants.
DOI: https://doi.org/10.7554/eLife.31065.028

• Supplementary file 2. Cq values of genes assayed by qPCR in sibling tail samples. The Cq data for *Figure 4—figure supplement 1* is shown.
DOI: https://doi.org/10.7554/eLife.31065.029

• Supplemental file 3. Primers used to make in situ hybridization probes
DOI: https://doi.org/10.7554/eLife.31065.030

• Supplemental file 4. Primers for qPCR
DOI: https://doi.org/10.7554/eLife.31065.031

• Transparent reporting form
DOI: https://doi.org/10.7554/eLife.31065.032

## Major datasets

The following dataset was generated:

| Author(s) | Year | Dataset title | Dataset URL | Database, license, and accessibility information |
|---|---|---|---|---|
| David Kimelman, Natalie L Smith, Jason Kuan Han Lai, Didier YR Stainier | 2017 | Zebrafish yap1/wwtr1 mutants | http://www.ncbi.nlm.nih.gov/geo/query/acc.cgi?acc=GSE102606 | Publicly available at the NCBI Gene Expression Omnibus (accession no. GSE102606) |

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
