## [Decision Letter]

Thank you for choosing to send your work, "Regulation of posterior body and ectodermal morphogenesis in zebrafish by localized Yap1 and Wwtr1", for consideration at *eLife*. Your article has been reviewed by three peer reviewers, and the evaluation has been overseen by a Senior/Reviewing Editor.

While all of the reviewers are enthusiastic about the study, they also agree that major revisions are necessary to bring this to publishable form. In particular, additional data are needed to support the conclusion that the defects are due to fibronectin misregulation. Genetic mosaic experiments are needed to determine whether the fin defect is due to a migration defect. Also, more detailed presentation of the expression profiling data is in order. Included below are the detailed comments of all three reviewers for your reference.

*Reviewer #1:*

The authors present the analysis of double mutant zebrafish embryos for *yap* and *wwrt1*. This represents one of the first substantial analyses of the role of Yap in the embryonic development of zebrafish and represents an important contribution to the field.

Despite opting for a genome wide analysis of the genes that are mis-regulated upon mutation in *yap^-/-^;wwtr1^-/-^*mutants, only a very limited analysis of this dataset is performed in the manuscript. At the very least it would be of interest to look into the top genes that are upregulated as well as downregulated. Although, it is not clear from Table 1 whether the 'fold change' that is referred to is positive or negative. In addition, there are a number of other common targets for *yap* that have been found in other contexts (such as CTGF), it would be useful to see whether these are also modulated in the context of the zebrafish. I think that these analyses are necessary to support the conclusions made about notochord and ectodermal markers being the predominant sets of genes that we mis-regulated in the mutants.

In order to conclude about the role of the notochord phenotype in generating defects in posterior body development, it is necessary to include a more detailed comparison to the *yap-/-;wwtr1-/-* mutants. While the text states that the defects paralleled those of the *yap-/-;wwtr1-/-* mutant, it is not clear what aspects of the phenotype are the same or different. For example, the timing of somitogenesis and overall somite number, as well at the timing of axial truncation could easily be determined for both mutants and displayed alongside one another. As it stands, I am not sure that anything substantial is gained in this comparison. Furthermore, the fact that defects in *yap^-/-^;wwtr1^-/-^*mutants precede those in embryos where the notochord is absent suggest that there is an additional function of *yap/wwtr1* at earlier stages that precede both its role in the notochord and in fin formation. The authors could make use of their heat-shock inducible lines in order to temporally separate these events.

Similarly, a more detailed spatiotemporal analysis of the onset of disruptions in the Fn deficient embryos in comparison to the *yap^-/-^;wwtr1^-/-^*mutants is required in order to conclude that this is a phenocopy.

The authors claim on the basis of their live imaging experiments that the cells forming the ventral fin fold are 'highly migratory' and that these behaviours are moderately reduced in the mutants. It is not clear from the one control movie shown in the paper whether the cells are migratory, as it seems that only very little cell rearrangement is occurring. While the cells are undergoing some cell shape changes, this is not quantified or correlated with the formation of the fin fold in time and space. In order to attribute specific cell behaviours to fin fold formation, it is necessary to quantify these behaviours and compare them to ectodermal cells in regions that do not form a fin, or at stages preceding fin formation. These same parameters can be then compared to the mutant situation. At the very least the authors should check cell velocity, directionality and alterations in cell shape in order to support their conclusions. Also, the number of videos that have been analysed in each condition should be detailed.

*Reviewer #2:*

The study by Kimelman et al., addresses the timely question whether and how Yap signaling is involved in vertebrate body elongation. In recent years, Yap has been implicated in mechanotransduction. While a number of recent studies have examined the role of mechanics in body elongation, none have address the obvious question of how Yap signaling may operate during elongation. The results are surprising in that Yap signaling is required for normal zebrafish body elongation but that Yap's function is restricted to the notochord and epidermis. Moreover, Yap does not appear to function by regulated *arhgaps*, as suggested by studies in organoids, but rather perturbs the Fibronectin matrix. The current study is thorough, and includes the generation of zebrafish knockouts for Yap, its paralog *wwtr1*, and characterization of double mutants. Transcriptional profiling of dissected tailbuds is used to characterize changes in gene regulation. Heat shock dominant negative and constitutively active Yap transgenes provide corroborating evidence. Overall, the study is very interesting, the manuscript is clearly written and, in general, the conclusions are supported by the data presented. However, there are a few conclusions that need to be better substantiated, as detailed below.

Expression of the dn-Fn construct approximates the *yap1;wwtr1* double mutant phenotype, but the authors should perform timelapse analysis of the dn-Fn fin fold morphogenesis to confirm that the phenotype is consistent at the level of cell behavior. The phenotypes could otherwise arise from slightly different mechanisms.

The proposition that adhesion of the ectoderm is reduced in the *yap1;wwtr1* double mutant needs to be strengthened. The authors are correct that fixation/over-fixation may obscure this phenotype. It could be revealing to image live *yap1;wwtr1* double mutants expressing a membrane GFP/ and nuclear RFP. In transverse sections, one would predict that there would be evident separation of the ectoderm from the dorsal somite.

The major question that comes to mind, particularly with respect to the ectoderm phenotype, is whether the phenotype is due to failure to create a normal Fibronectin matrix, which is then affecting cell migration, or whether the cells migrate abnormally and this affects the morphology of the Fibronectin matrix? Genetic mosaics could potentially address this question.

*Reviewer #3:*

The authors confirm previous findings from Medaka indicating that Yap and Wwtr1 are essential for posterior axis elongation. Whereas, previously it has been suggested that axis elongation in Yap/Wwtr1 loss of function is caused by an inability to combat gravity these authors suggest a role for Yap and Wwtr1 in ectodermal and notochord morphogenesis. They show that Yap and Wwrt1 are expressed specifically in tailbud ectoderm and notochord. Moreover, through RNAseq analysis they find that YAP and Wwrt1 control the expression of exrg4b and wu:fc23c09 in the tailbud. In contrast, Yap and Wwrt1 do not control *arhgap12a/27* expression as previously reported. The authors then investigate ectodermal morphogenesis in mutants and observe a loss of ectodermal thickening associated with dorsal fin formation. Through live imaging the authors note that, in WT embryos, ectodermal cells appear to migrate toward the midline. In contrast, mutant ectodermal cells do not accumulate in the midline despite behaving dynamically. As Fibronectin is an important factor in axis elongation and tissue adhesion the authors analyze FN deposition and find that FN is not maintained in newly forming somites where *yap* and *wwrt1* is expressed. Finally, they indicate that epithelial integrity is, therefore, lost in mutants.

This article is appropriate for publication in *eLife* with major revisions.

Fin fold formation is thought to occur either through shape changes in static cells or movement of the whole epidermal sheet toward the midline. The authors attempt to clarify this controversy through live imaging. The authors find cells that enter the midline over time and determine that this occurs through cell migration. However, these data do not exclude that convergent extension like movements could be sufficient to move cells toward the midline. Here, the appearance of cells in the midline could be mistaken for cell migration when their appearance is actually the consequence of a cell rearrangement driven, perhaps, through t1 transitions. Their conclusions are therefore overstated and should be experimentally addressed for publication.

Although their data are highly correlative with a role for FN downstream of Yap and *wwrt1*, these data do not confirm this interaction. An epistasis experiment is required to determine whether FN is downstream of Yap and Wwrt1.

---

## [Author Response]

Reviewer #1:

*[…] Despite opting for a genome wide analysis of the genes that are mis-regulated upon mutation in yap^-/-^;wwtr1^-/-^mutants, only a very limited analysis of this dataset is performed in the manuscript. At the very least it would be of interest to look into the top genes that are upregulated as well as downregulated. Although, it is not clear from Table 1 whether the 'fold change' that is referred to is positive or negative. In addition, there are a number of other common targets for yap that have been found in other contexts (such as CTGF), it would be useful to see whether these are also modulated in the context of the zebrafish. I think that these analyses are necessary to support the conclusions made about notochord and ectodermal markers being the predominant sets of genes that we mis-regulated in the mutants.*

The reviewer felt that the data analysis was very limited and we agree and so we have done a number of things to improve the analysis:

1) We have done more in situ analysis and now examine in Table 1 the expression of the top 12 genes that change in the RNA-seq dataset, not just the top 7. All are in the presumptive epidermis (a term we are now using in place of ectoderm) or notochord, as in our previous conclusion.

2) As the reviewer suggested, we collated known target Yap1/Wwtr1 genes from two large scale analyses in other systems and present this information in Table 2. All except one of these known Yap targets are expressed in the presumptive epidermis or notochord. The one exception is one weakly expressed gene in the ventral mesoderm, which although we did not obviously detect Yap1/Wwtr1 in the ventral mesoderm, suggests that it regulates a very limited subset of genes in this tissue.

3) We have now added a new figure (Figure 4) that provides Gene Ontology analysis of all of the genes that we see significantly changed in the *yap1;wwtr1* mutants (as provided in Supplementary file 1). Interestingly, Biological Process analysis (Figure 4) identifies genes in the epidermis or fin (the fin is primarily an epidermal structure), again supporting our idea that the prospective epidermis is a key target tissue.

4) Because the changes in known Yap1/Wwtr1 genes (in Table 2) are more modest than our top hits, we validated the changes in the Table 2 genes by qPCR and provide that data in Figure 4—figure supplement 1.

5) We now make clear that the changes we see are all decreases in expression.

We appreciate the reviewer pushing us to extend the data analysis.

*In order to conclude about the role of the notochord phenotype in generating defects in posterior body development, it is necessary to include a more detailed comparison to the yap^-/-^;wwtr1^-/-^mutants. While the text states that the defects paralleled those of the yap^-/-^;wwtr1^-/-^mutant, it is not clear what aspects of the phenotype are the same or different. For example, the timing of somitogenesis and overall somite number, as well at the timing of axial truncation could easily be determined for both mutants and displayed alongside one another. As it stands, I am not sure that anything substantial is gained in this comparison. Furthermore, the fact that defects in yap^-/-^;wwtr1^-/-^mutants precede those in embryos where the notochord is absent suggest that there is an additional function of yap/wwtr1 at earlier stages that precede both its role in the notochord and in fin formation. The authors could make use of their heat-shock inducible lines in order to temporally separate these events.*

We had shown the notochord mutant mostly to make the point that the *yap1;wwtr1* double mutant phenotype could not be explained only by a defect in the notochord, but the way we presented it made it seem like we were claiming more than we intended.

We have removed the figure with the *noto* mutant in accordance with the reviewer’s opinion, and in the text just make it clear that even an embryo with no notochord does not have severe defects as seen in the *yap1;wwtr1* double mutants, which explains why we also focused on the presumptive epidermis. We also don’t raise it in the text, but we have done careful examination of somite number between controls and mutants and see no differences. Since this is essentially a negative result we have not included it as we feel it would distract from the paper. Finally, as a note of clarity, we do see defects in the *noto* mutants at the same time (~16 somites) as when we see defects in the *yap1;wwtr1* double mutants so we don’t think that *yap1;wwtr1* has an earlier role than seen in *noto* mutants, just a much more severe role.

*Similarly, a more detailed spatiotemporal analysis of the onset of disruptions in the Fn deficient embryos in comparison to the yap^-/-^;wwtr1^-/-^mutants is required in order to conclude that this is a phenocopy.*

It was a mistake on our part to use the word phenocopy. What we intended to say was that the Fn deficient embryos that make it through gastrulation appear remarkably similar to the *yap1;wwtr1* double mutants. Because mRNA injection is quite variable, a number of the embryos exhibit gastrulation defects because of a role for Fn in gastrulation but survivors have a clear morphogenetic defect. We now show pictures of the whole embryo (Figure 10), not just the posterior as in the initial version of the manuscript, to make the comparison to the *yap1;wwtr1* double mutants clearer.

The authors claim on the basis of their live imaging experiments that the cells forming the ventral fin fold are 'highly migratory' and that these behaviours are moderately reduced in the mutants. It is not clear from the one control movie shown in the paper whether the cells are migratory, as it seems that only very little cell rearrangement is occurring. While the cells are undergoing some cell shape changes, this is not quantified or correlated with the formation of the fin fold in time and space. In order to attribute specific cell behaviours to fin fold formation, it is necessary to quantify these behaviours and compare them to ectodermal cells in regions that do not form a fin, or at stages preceding fin formation. These same parameters can be then compared to the mutant situation. At the very least the authors should check cell velocity, directionality and alterations in cell shape in order to support their conclusions. Also, the number of videos that have been analysed in each condition should be detailed.

We agree with the reviewer and thank him/her for this suggestion. In the revised version we now provide measurements of speed, as well as the time spent at the midline, and changes in cell shape. What is very interesting and that has emerged from this data is that not only are the wild-type presumptive epidermal cells migrating, but that the mutant cells are actually moving faster. In addition, whereas the wild-type cells move into the midline and stay there, the mutant cells move to the midline and then cross to the other side, which is why the fin fold fails to form. Finally, whereas the wild- type cells become square shaped over time, the mutant cells retain the elongated (migratory) morphology. All of this data is now presented in new Figure 8. We did, as the reviewer suggests, examine the lateral surfaces of the embryo and do see the presumptive epidermal cells moving there as well. Thus, there is an overall movement of presumptive epidermal cells both dorsally and ventrally, causing the accumulation at the dorsal and ventral surfaces with the resulting extension of the fin folds. We now call this concerted migration since the wild-type cells are moving from both sides to the midline and pushing against each other to cause the fin to extend dorsally or ventrally, with a concomitant shape change in the epidermal cells. In contrast, in the mutants, with defective Fn, the cells are just moving around but cannot create the forces to form the fin fold, and so they transit across the midline and do not undergo cell shape changes. We have incorporated all this information into the revised manuscript. The major point we want to make is that the formation of the fin fold is not just a static bending process as had previously been assumed, but involves actively migrating cells.

The number of videos examined is now provided in the legend.

Reviewer #2:[…] Expression of the dn-Fn construct approximates the yap1;wwtr1 double mutant phenotype, but the authors should perform timelapse analysis of the dn-Fn fin fold morphogenesis to confirm that the phenotype is consistent at the level of cell behavior. The phenotypes could otherwise arise from slightly different mechanisms.

The reviewer makes a very good point about the dn-Fn. Accordingly, we have now filmed dn-Fn embryos to compare the cell behavior. Because the dn-Fn is injected as mRNA, the results are more variable because the mRNA is not distributed equally among all cells in zebrafish embryos. Nonetheless, we see similar trends as in the mutants, with cells transiting through the midline, some cells moving faster than wild-type ones (and none moving slower), and cells having a longer height-length ratio, as with the mutants. We present the new speed and cell dimension data in a new Figure 8, which we feel helps strengthen the similarity between the mutant and Fn∆C injected cells. We appreciate the recommendation to examine this.

The proposition that adhesion of the ectoderm is reduced in the yap1;wwtr1 double mutant needs to be strengthened. The authors are correct that fixation/over-fixation may obscure this phenotype. It could be revealing to image live yap1;wwtr1 double mutants expressing a membrane GFP/ and nuclear RFP. In transverse sections, one would predict that there would be evident separation of the ectoderm from the dorsal somite.

We have followed the reviewer’s suggestion and live imaged embryonic development in wild-type and mutants using a membrane tagged GFP (we also had a nuclear RFP as suggested but the data is easier to see with just the GFP). Just as the reviewer predicted, we see a very clear separation of the presumptive epidermis from the somites as the severity of the overall mutant phenotype increases. We think this is a much better result than the data we had in the initial manuscript with lightly fixed embryos, so we have replaced the original Figure 10 with this new result (which is Figure 11 since we added a new Figure 8). We greatly appreciate the reviewer’s suggestion that we pursue this experiment.

The major question that comes to mind, particularly with respect to the ectoderm phenotype, is whether the phenotype is due to failure to create a normal Fibronectin matrix, which is then affecting cell migration, or whether the cells migrate abnormally and this affects the morphology of the Fibronectin matrix? Genetic mosaics could potentially address this question.

The reviewer raises an interesting question as to whether cell migration is affecting the Fn assembly. Because Fn is secreted by all cells, and because the cells are moving (including presumptive epidermis and somites) it would be very difficult to get a convincing result using genetic mosaics. However, we found that treating embryos with 40 μm Blebbistatin at the 12-somite stage prevents movement of the cells and consequently prevents formation of the fin fold, recapitulating the *yap1;wwtr1* double mutant phenotype. We then examined Fn assembly in the Blebbistatin treated embryos. Interestingly, while the Fn staining in the boundaries was completely absent, the staining under the epidermis (and around the notochord) was completely intact. Thus, it is the absence of Yap1 and Wwtr1 that is affecting the Fn assembly and not the aberrant migration of the cells. This is an interesting result and we have added it to the Discussion subsection “Presumptive epidermis morphogenesis in *yap1;wwtr1* double mutants”.

Reviewer #3:[…] This article is appropriate for publication in eLife with major revisions.Fin fold formation is thought to occur either through shape changes in static cells or movement of the whole epidermal sheet toward the midline. The authors attempt to clarify this controversy through live imaging. The authors find cells that enter the midline over time and determine that this occurs through cell migration. However, these data do not exclude that convergent extension like movements could be sufficient to move cells toward the midline. Here, the appearance of cells in the midline could be mistaken for cell migration when their appearance is actually the consequence of a cell rearrangement driven, perhaps, through t1 transitions. Their conclusions are therefore overstated and should be experimentally addressed for publication.

We appreciate the reviewer’s concern. We have accordingly done a lot more filming to look at this issue. Filming is quite tricky in this system compared to the beautiful studies done in *Drosophila* since the whole embryo is changing shape during this period as it is extending along the A-P axis. Despite this, we have looked very carefully and not seen anything that looks like the t1 transitions as seen in *Drosophila* (there are a lot of cell movements and shape changes, but they do not follow any typical order). We also do not believe that this is convergence-extension behavior since we see no evidence of mediolateral intercalations. Instead, we see a gradual movement of new cells into the fin fold, and we now provide measurement of the speed of their movement (new Figure 8). We also now show (new Figure 8) that whereas the wild-type cells move into the fin fold and stay there, mutant cells transit through the fin fold. If this was a convergence extension defect we would not expect this behavior and instead expect that while the fin fold would not form, cells at the midline would stay in place.

We also showed our data to an expert in epidermal morphogenesis (Celeste Berg), who said it was unlike what is seen in other systems, including *Drosophila*. Our new interpretation is that the cells have an inherent migratory capacity as seen when Fn is disrupted (either in the *yap1;wwtr1* double mutants or with the dom-neg Fn); but when Fn is assembled normally, the cells moving to the midline from the left and right sides push against each other, thus limiting their movement, causing them to change shape to a much more square morphology, and thus to extend the fin fold. We have included this new data in the revised manuscript and appreciate the reviewer encouraging us to provide a better description of cell movements.

Finally, one of our main purposes in describing in some detail the epidermal morphogenesis of the fin fold was to get researchers to understand that the fin fold does not just form from the bending of an epidermal sheet as has long been thought, but to realize it is a much more dynamic process, with very interesting properties. Our hope is that our description will catalyze more studies in this area that will further refine the exact movements involved in this process.

Although their data are highly correlative with a role for FN downstream of Yap and wwrt1, these data do not confirm this interaction. An epistasis experiment is required to determine whether FN is downstream of Yap and Wwrt1.

The reviewer raises an interesting point. Since Yap1 and Wwtr1 are mechanosensors, not only could a lack of Yap1/Wwtr1 cause changes in Fn as we showed (i.e. Fn is epistatic to Yap1/Wwtr1) but changes in Fn could alter Yap1/Wwtr1 (i.e. Yap1/Wwtr1 is epistatic to Fn). To test this, we examined Yap1 using our Yap1 antibodies in embryos expressing the dominant-negative Fn compared to uninjected embryos. We found that the expression of Yap1 was unaffected by disruptions in Fn, and thus we can say that Yap1 is not epistatic to Fn. We appreciate the reviewer raising this point and have added this to the last paragraph of the subsection “Yap1/Wwtr1-dependent Fibronectin assembly is required for body extension and morphogenesis of the presumptive epidermis”.